# Unexpected plasticity in the life cycle of *Trypanosoma brucei*

**Sarah Schuster[1†], Jaime Lisack[1†], Ines Subota[1†], Henriette Zimmermann[1], Christian Reuter[1], Tobias Mueller[2], Brooke Morriswood[1], Markus Engstler[1]\***

[1]Lehrstuhl für Zell- und Entwicklungsbiologie, Biozentrum, Julius-Maximilians-Universität, Würzburg, Germany; [2]Lehrstuhl für Bioinformatik, Biozentrum, Julius-Maximilians-Universität, Würzburg, Germany

**Abstract** African trypanosomes cause sleeping sickness in humans and nagana in cattle. These unicellular parasites are transmitted by the bloodsucking tsetse fly. In the mammalian host's circulation, proliferating slender stage cells differentiate into cell cycle-arrested stumpy stage cells when they reach high population densities. This stage transition is thought to fulfil two main functions: first, it auto-regulates the parasite load in the host; second, the stumpy stage is regarded as the only stage capable of successful vector transmission. Here, we show that proliferating slender stage trypanosomes express the mRNA and protein of a known stumpy stage marker, complete the complex life cycle in the fly as successfully as the stumpy stage, and require only a single parasite for productive infection. These findings suggest a reassessment of the traditional view of the trypanosome life cycle. They may also provide a solution to a long-lasting paradox, namely the successful transmission of parasites in chronic infections, despite low parasitemia.

**\*For correspondence:**
markus.engstler@biozentrum.uni-wuerzburg.de

[†]These authors contributed equally to this work

**Competing interests:** The authors declare that no competing interests exist.

## Introduction

Trypanosomes are among the most successful parasites. These flagellated protists infect all vertebrate classes, from fish to mammals, and can cause devastating diseases. African trypanosomes, which are transmitted by the tsetse fly, are the agents of nagana in livestock and sleeping sickness in humans (*Bruce, 1895*). The most intensively s tudied African trypanosome subspecies is *Trypanosoma brucei brucei,* which in the past decades has emerged as a genetic and cell biological model parasite. The life cycle of *T. brucei* was initially elucidated more than a century ago. As part of their life cycle, the trypanosomes undergo a full developmental program in the tsetse fly in order to become infective to the mammalian host (*Koch, 1909*). This finding, made by Kleine in 1909, showed that transmission was not a purely mechanical event (*Kleine, 1909*). Kleine subsequently found that the life cycle in the fly could take up to several weeks to complete, a discovery that was shortly afterwards confirmed by *Bruce et al., 1909*. More details of the general life cycle of *Trypanosoma brucei* were then elucidated by Robertson in 1913, with several key observations concerning the transmission event (*Robertson and Bradford, 1913*). Subsequent work has resulted in a detailed picture of the passage through the fly, beginning with the ingestion of trypanosomes in an infected bloodmeal (*Rotureau and Van Den Abbeele, 2013*). After entering through the tsetse proboscis, the infected blood is either held for a short time in the crop, which acts as a storage site and allows the tsetse to drink more blood per meal, or is passed directly to the midgut. Upon entering the tsetse midgut, the trypanosomes differentiate into the proliferative procyclic stage. Once established in the midgut, the parasites must pass the peritrophic matrix, a protective sleeve that separates the bloodmeal from midgut tissue. To do this, the parasites are thought to swim up the endotrophic space to the proventriculus, the site of peritrophic matrix synthesis, where they are able to cross to the ectotrophic space (*Rose et al., 2020*). After having crossed the peritrophic matrix and entered

the ectroperitrophic space, procyclic trypanosomes may either further colonise the ectotrophic anterior midgut, becoming the cell-cycle arrested mesocyclic stage, or continue directly to the proventriculus. In the proventriculus, trypanosomes further develop into the long, proliferative epimastigote stage (*Rose et al., 2020*). The epimastigotes then swim from the proventriculus to the salivary glands, while undergoing an asymmetric division to generate a long and a short daughter cell. Once in the salivary gland, the long daughter cell is thought to die while the small one attaches via its flagellum to the salivary gland epithelium (*Vickerman, 1969*). The attached epimastigotes are proliferative, producing either more attached epimastigote daughter cells or freely swimming, cell cycle-arrested metacyclic trypanosomes. As early as 1911, it was clear that the metacyclic stage (at that time called metatrypanosomes) is the only mammalian-infective stage (*Bruce et al., 1911*).

In the mammalian host, trypanosomes have been found in many different organs, including brain, skin, and fat, but are hard to study experimentally (*Capewell et al., 2016*; *Goodwin, 1970*; *Krüger et al., 2018*; *Trindade et al., 2016*). The two main stages found in the bloodstream, and the best-characterised experimentally, are the proliferating slender bloodstream stage and the cell cycle-arrested stumpy bloodstream stage (*Krüger et al., 2018*; *Matthews et al., 2004*; *Vickerman, 1985*). The stumpy stage is formed in response to quorum sensing of the stumpy induction factor (SIF), a signal produced by slender bloodstream trypanosomes (*Vassella et al., 1997*). As the stumpy stage only survives for 2–3 days after formation, the generation of stumpy parasites is thought to control the burden the parasites impose on the host (*Turner et al., 1995*). The SIF pathway that controls the slender-to-stumpy transition has been described down to the molecular level, with the *protein associated with differentiation* (PAD1) being the first recognised molecular marker for the stumpy stage trypanosomes (*Dean et al., 2009*; *Mony and Matthews, 2015*). More recently, it was also shown that the stumpy pathway can be triggered independently of SIF, although the extent to which this occurs in the general population remains unclear (*Batram et al., 2014*; *Zimmermann et al., 2017*). Besides its proposed role in controlling parasitaemia in the mammalian host, the stumpy stage has a second essential function in the trypanosome life cycle: it is believed to be the only life cycle stage that can infect the tsetse fly (*Rico et al., 2013*). Thus, arrest of the cell cycle and differentiation to the stumpy stage are presumed essential for developmental progression to the procyclic insect stage. As early as 1912, Robertson suggested that the short, stumpy bloodstream trypanosomes represent the fly-infective stage (*Robertson, 1912*). While this assumption was questioned several times throughout the 20th century, the discovery of quorum sensing and SIF in the 1990s made it become generally accepted (*Vassella et al., 1997*). However, if stumpy trypanosomes are the only stage that can infect the fly, another problem arises. Although trypanosomes might be found at higher densities in the skin (*Capewell et al., 2016*), chronic trypanosome infections are characterised by low blood parasitemia, meaning that the chance of a tsetse fly ingesting any trypanosomes, let alone short-lived stumpy ones, is also very low (*Frezil, 1971*; *Wombou Toukam et al., 2011*). Mathematical models have been developed that aim to explain how the limited number of short-lived stumpy cells in the host blood and interstitial fluids can guarantee the infection of the tsetse fly, which is essential for the survival of the species (*Capewell et al., 2019*; *MacGregor and Matthews, 2008*; *Seed and Black, 1999*). The present study provides surprising new solutions to this problem. First, systematic quantification of infection efficiencies showed that very few trypanosomes are necessary to infect a tsetse fly, and in fact just one is sufficient. Second, and wholly unexpectedly, slender stages proved at least as competent at infecting flies as stumpy stages. These findings suggest greater plasticity in the life cycle than supposed, prompting a reassessment of the current rigid view of the process.

## Results

### A single trypanosome is sufficient for infection of a tsetse fly

Slender and stumpy bloodstream stage trypanosomes can be distinguished based on cell cycle, morphological, and metabolic criteria. The genome of the single mitochondrion (kinetoplast, K) and the cell nucleus (N) can be readily visualised using DNA stains, and their prescribed sequence of replication (1K1N, 2K1N, 2K2N) allows cell cycle stage to be inferred (*Sherwin and Gull, 1989*). Slender cells are found in all three K/N ratios, while stumpy cells, which are cell cycle-arrested, are found only as 1K1N cells (*Figure 1A*). Expression of the *protein associated with differentiation 1* (PAD1) is

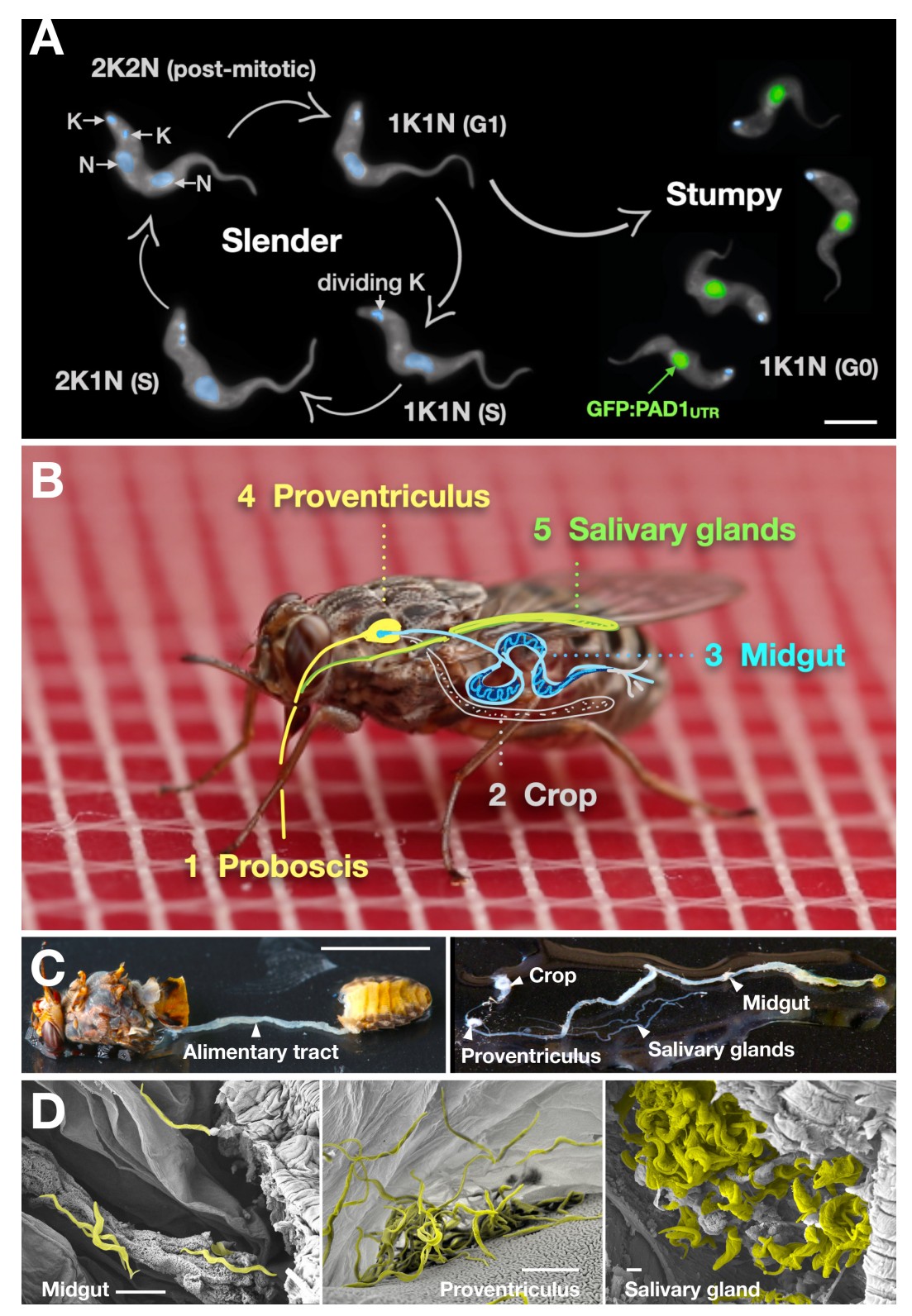

**Figure 1.** Slender trypanosomes can complete the entire life cycle in the tsetse fly vector. (A) Cell cycle (G1/S/post-mitotic), morphology, and differentiation of bloodstream form (mammalian-infective stage) trypanosomes. Proliferation of slender trypanosomes is detectable by duplication and segregation of the mitochondrial genome (kinetoplast, K) and nuclear DNA (N) over time. Quorum sensing causes cell cycle arrest (G0) and expression of the stumpy marker PAD1. Images are false-coloured, maximum intensity projections of deconvolved 3D stacks. The green colour indicates the

*Figure 1 continued on next page*

*Figure 1 continued*

nuclear GFP:PAD1$^{UTR}$ fluorescence, the DAPI-stained kinetoplast and nucleus are shown in light blue, and the AMCA-sulfo-NHS-labelled parasite cell surface are shown in grey. Scale bar: 5 µm. (B) Trypanosome infections of tsetse flies were achieved via bloodmeal, which consists typically of 20 µl, through a silicon membrane. To complete infection in a tsetse fly after an infective blood-meal, trypanosomes first travel to the midgut, followed by the proventriculus, and finally must reach the salivary glands. (C) The first panel depicts a dissected tsetse fly for explantation of the alimentary tract. The second panel shows the explanted alimentary tract of the tsetse, with the different subcompartments labelled. Scale bar: 5 mm. (D) Scanning electron micrograph of a typical trypanosome infection of the tsetse midgut, proventriculus, and salivary glands. Parasites are false-coloured yellow. Scale bar: 1 µm.

The online version of this article includes the following video and figure supplement(s) for figure 1:

**Figure supplement 1.** Stumpy trypanosomes express PAD1 on their surface when the GFP:PAD1$^{UTR}$ reporter is expressed.

**Figure 1—video 1.** Video of a tsetse fly taking a bloodmeal through a silicone membrane.

https://elifesciences.org/articles/66028#fig1video1

accepted as a marker for development to the stumpy stage (*Dean et al., 2009*). As the 3'UTR of the PAD1 gene regulates the expression of *pad1* (*MacGregor and Matthews, 2012*), cells expressing an NLS-GFP reporter fused to the 3' UTR of the PAD1 gene (GFP:PAD1$^{UTR}$) will have GFP-positive nuclei when the PAD1 gene is active. Hence, slender cells are GFP-negative; stumpy cells are GFP-positive (*Figure 1A*). The validity of the GFP:PAD1$^{UTR}$ reporter as an indicator for the activation of the PAD1 pathway has been reported previously (*Batram et al., 2014*; *Zimmermann et al., 2017*), and was further corroborated by co-staining with an antibody against the PAD1 protein (*Figure 1— figure supplement 1*). We have previously shown that stumpy cells can be formed independently of high cell population density by ectopic expression of a second variant surface glycoprotein (VSG) iso-form, a process that mimics one of the pathways involved in trypanosome antigenic variation (*Batram et al., 2014*; *Cross, 1975*; *Hertz-Fowler et al., 2008*; *Zimmermann et al., 2017*). These so-called expression site (ES)-attenuated stumpy cells can complete the developmental cycle in the tsetse fly (*Zimmermann et al., 2017*). It remained an open question whether this occurred with the same efficiency as with SIF-produced stumpy cells. Therefore, we quantitatively compared the trans-mission competence of stumpy populations generated by either SIF treatment or through ES-attenu-ation. Tsetse flies (*Glossina morsitans morsitans*) were infected via membrane feeding (*Figure 1B*; *Figure 1—video 1*) with defined numbers of pleomorphic stumpy trypanosomes, capable of com-pleting the entire developmental cycle. This cycle includes entrance through the proboscis, passage through the crop, then establishing infections in the midgut, proventriculus, and finally the salivary glands (*Figure 1B*). Two transgenic trypanosome cell lines, both of which contained the GFP: PAD1$^{UTR}$ reporter construct, were used. One was subjected to tetracycline-induced, ectopic VSG expression to drive ES attenuation (*Figure 2A*, lines i-iii, Stumpy$^{ES}$) (*Zimmermann et al., 2017*). The other was treated with stumpy induction factor (SIF) (*Figure 2A*, rows iv-vi, Stumpy$^{SIF}$). Both treat-ments resulted in expression of the GFP:PAD1$^{UTR}$ reporter and rapid differentiation to the stumpy stage. The resulting stumpy populations were fed to tsetse flies at concentrations ranging from 120,000 to 10 cells/ml. A feeding tsetse typically ingests about 20 µl of blood (*Gibson and Bailey, 2003*), meaning that, on average, between 2400 and 0.2 trypanosomes were ingested per blood-meal (*Figure 2A*, rows i-vi, column 2, Total; *Figure 2—figure supplement 1*). The trypanosomes had previously been scored for expression of the GFP:PAD1$^{UTR}$ reporter to confirm their identity as the stumpy stage (*Figure 2A*, columns 3–4). To analyse the infections, we carried out microscopic analyses of explanted tsetse digestive tracts (*Figure 1C*). The dissection of the flies was done 5–6 weeks post-infection. The presence of mammal-infective, metacyclic trypanosomes in explanted tse-tse salivary glands indicated the completion of the life cycle inside the tsetse. Remarkably, the uptake, on average, of just two stumpy parasites of either cell line produced robust infections of tse-tse midgut (MG), proventriculus (PV), and salivary glands (SG) (*Figure 1D*; *Figure 2A*, columns 5–7, *Figure 2B*, *Figure 2—figure supplement 2*). Ingestion, on average, of even a single stumpy cell was sufficient to produce salivary gland infections in almost 5% of all tsetse (*Figure 2A*, row v). When the stumpy parasite number was further reduced to 0.2 cells on average per bloodmeal, meaning only every 5th fly would receive a stumpy cell, 0.9% of flies still acquired salivary gland infections (*Figure 2A*, row vi). As a measure of the incidence of life cycle completion in the tsetse fly, we calcu-lated the transmission index (TI) for each condition. The TI has been previously defined as the ratio of salivary gland to midgut infections and hence, it is a measure for successful passage through the

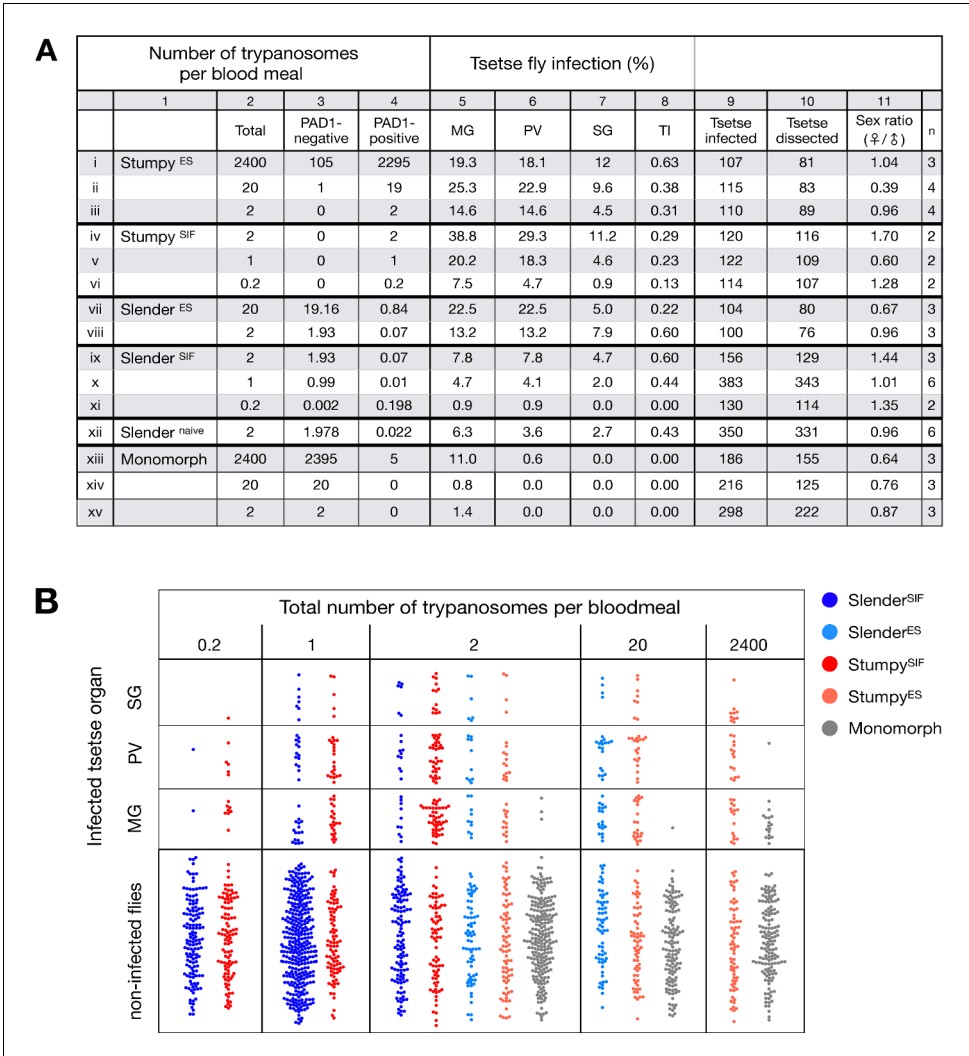

**Figure 2.** Slender trypanosomes can complete the entire tsetse infection cycle, and a single parasite is sufficient for tsetse passage. The flies were infected with different numbers of either stumpy or slender trypanosomes. Slender[ES], Slender[SIF], and monomorphic trypanosome cell lines were cultivated with regular dilution and a maximum population density of $5 \times 10^5$ cells/ml, in order to avoid SIF accumulation. Stumpy development was induced by expression site attenuation (ES), or SIF-treatment (SIF). Stumpy[ES] induction was performed by ectopic overexpression of VSG121 for 56 hr, in the absence of SIF. SIF-mediated stumpy transition (Stumpy[SIF]) was induced by incubating slender trypanosome populations in the presence of SIF-containing conditioned medium for 48 hr. The expression of the stumpy marker PAD1 was checked before fly feeding. (**A**) Percent of the total fly infections is shown. MG, midgut infection; PV, proventriculus infection; SG, salivary gland infection; TI, transmission index (number of SG infections divided by MG infections); n, number of independent fly infections. (**B**) Graphical visualisation (beeswarm plot) of the data shown in panel A, colour-coded according to cell population used. MG, midgut; PV, proventriculus; SG, salivary gland; n, number of independent fly infection experiments.

The online version of this article includes the following figure supplement(s) for figure 2:

**Figure supplement 1.** Number of trypanosomes per milliliter (ml) of blood correlating to how many trypanosomes would be found in a tsetse bloodmeal.

**Figure supplement 2.** Slender trypanosomes can complete the entire tsetse infection cycle, and a single parasite is sufficient for tsetse passage.

second part of the trypanosome tsetse cycle, where trypanosomes are again infective to a mammalian host (*Figures 1B*, 3-5)(*Peacock et al., 2012*). We found that for flies infected with two trypanosomes on average, the TI was comparable between SIF-induced (TI = 0.29) and ES-induced (TI = 0.31) stumpy trypanosomes (*Figure 2A*, rows iii-iv; *Figure 3*). A similar TI of 0.23 was observed in

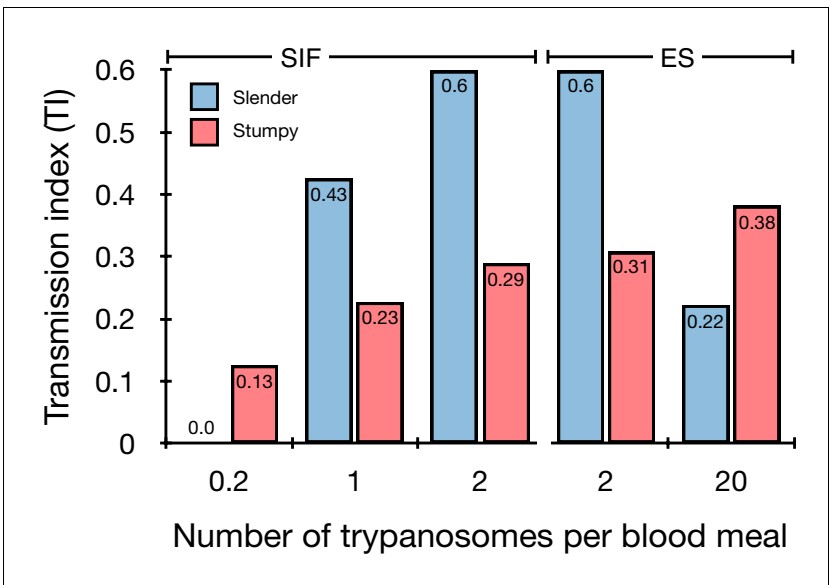

**Figure 3.** Graphical representation of the transmission index TI (number of salivary gland infections divided by the number of midgut infections) of slender (blue) and stumpy (red) trypanosomes at different numbers of trypanosomes per bloodmeal (data reproduced from *Figure 2A*, column 8). A high TI indicates successful completion of the life cycle in the tsetse vector. At low infective doses, slender trypanosomes had a higher TI compared to stumpy parasites. There was no difference between stumpy parasites generated by SIF-treatment (SIF) or expression site attenuation (ES).

The online version of this article includes the following figure supplement(s) for figure 3:

**Figure supplement 1.** Graphical representation of the transmission index (TI - number of salivary gland infections divided by midgut infections) of slender (blue) and stumpy (red) trypanosomes for the individual experiments (n) at different numbers of trypanosomes per bloodmeal (individual replicates from data in *Figure 2A*, column 8).

**Figure supplement 2.** Transmission index TI (number of salivary gland infections divided by midgut infections) of slender and stumpy trypanosomes for the individual experiments (n) at different numbers of trypanosomes per bloodmeal (individual replicates from data in *Figure 2A*, column 8).

flies ingesting on average one trypanosome (*Figure 2A*, row v; *Figure 3*). Thus, our data not only clearly show that SIF- and ES-induced stumpy parasites are equally efficient in completing the weeks-long, multi-step fly cycle, but also that a single stumpy cell is sufficient to produce a mature fly infection (individual replicates can be seen in *Figure 3—figure supplement 1* and *Figure 3—figure supplement 2*). While this may seem comparable with an observation that has been made before for *Trypanosoma congolense* (*Maudlin and Welburn, 1989*), the migration through the fly differs between the two species: *T. brucei* infects the salivary glands, while *T. congolense* infects the proboscis. The tsetse fly, however, is much more susceptible to infections with *T. congolense* than with *T. brucei*, with a nearly 5-fold increase in percent *T. congolense* proboscis infections as compared to *T. brucei* salivary gland infections. As the authors of this work used chemicals (*L*- glutathione and *N*-acetylglucosamine) to boost *T. brucei* infection rates, the fivefold difference is actually a lower estimate (*Peacock et al., 2012*). Our results demonstrate that very low numbers of *T. brucei* stumpy cells can also successfully establish mature tsetse fly infections.

## Proliferating slender bloodstream stage trypanosomes infect the insect vector with comparable efficiency to cell cycle-arrested stumpy bloodstream stage parasites

Originally intended as a control experiment with an easily predictable (negative) outcome, we also infected tsetse flies with proliferating PAD1-negative slender trypanosomes from the two pleomorphic cell lines used (*Figure 2A*, rows vii-xi). Unexpectedly, we found that slender parasites were not only viable in the midgut, but also infected the proventriculus and the salivary glands. (*Figure 2A*, rows vii-xi; *Figure 2B*). Even one slender parasite was on average sufficient to establish midgut infections, proving that slender and stumpy parasites are, in principle, equally viable in the tsetse midgut.

The infection efficiency, measured using TI, was similar when the flies were fed with either 20 stumpy trypanosomes or 20 pleomorphic slender trypanosomes (*Figure 2A*, compare TI in column eight for rows ii and vii). When flies were fed with an average of two slender parasites each, the TI was actually higher for slender cells (0.60) than for stumpy cells (0.31) (*Figure 3*; *Figure 3—figure supplement 1*; *Figure 3—figure supplement 2*). This TI of 0.60 was identical for both populations of slender cells (*Figure 3*). Next, when given, on average, just one PAD1-negative slender cell per bloodmeal, parasite infections were still established in the midgut, proventriculus, and salivary glands with incidences of 4.7%, 4.1%, and 2.0% respectively, at a TI of 0.44 (*Figure 2A*, row x; *Figure 3*). In order to be absolutely sure that slender trypanosomes can passage through the tsetse, we repeated the experiment with naïve slender parasites that had been freshly differentiated from insect-derived metacyclic trypanosomes, that is cells that had just restarted the mammalian life cycle stage (*Figure 2A*, row xii). Infections with, on average, two freshly-differentiated slender trypanosomes per bloodmeal revealed 6.3% midgut and 2.7% salivary gland infections. The transmission index was 0.43 (*Figure 2A*. row xiii). This important control formally ruled out that cultivated slender cells had undergone any kind of gain-of-function adaptation in culture that made them transmission-competent.

As another control for the slender infection experiments, tsetse infections were carried out using a monomorphic slender trypanosome strain, that is one that had lost the capacity of differentiating to the stumpy stage (*Figure 2A*, rows xiii - xv). Monomorphic trypanosomes are able to infect the tsetse midgut, but they are incapable of establishing robust infections and completing the developmental cycle in the fly (*Herder et al., 2007*; *Peacock et al., 2008*). As expected, no salivary gland infections were seen using these cells, even at high infection numbers. Interestingly, we found that even two monomorphic slender parasites could establish a fly midgut infection (*Figure 2A*, row xv). Thus, infection of the tsetse midgut is independent of the capacity for developmental progression and the infective dose, and it does not require the stumpy life cycle stage. This finding also challenges the assumption that slender parasites are selectively eliminated from the parasite population and that only stumpy trypanosomes can survive the harsh conditions thought to prevail within the tsetse crop and midgut (*Nolan et al., 2000*).

The ES-attenuated cells showed similar midgut, proventriculus, and salivary gland infection incidence as either the stumpy or slender stage (*Figure 2A*, rows ii-iii and vii-viii, *Figure 2B*). The SIF-induced stumpy cells, however, appeared more effective in establishing midgut infections than their slender counterparts (*Figure 2A*, rows iv-vi and ix-xi, *Figure 2B*). This result could be interpreted as stumpy trypanosomes being more successful in the tsetse fly, but this is a conclusion that is clearly not supported by our data. First, the infections with one to two slender cells produced higher TI values than those with the same numbers of stumpy cells (*Figure 3*; *Figure 3—figure supplement 1*; *Figure 3—figure supplement 2*). This suggests that the proliferative slender cells are actually more capable of progressing from a midgut infection to a salivary gland one, and thus have at least comparable overall developmental competence to the stumpy stage. Second, the lack of correlation between infective dose and midgut infections underlines the importance of the TI as a relative measure. What is biologically relevant is not the initiation of infection but the completion of the tsetse passage. In summary, our experiments not only establish that a single *T. brucei* cell (either slender or stumpy) can infect the tsetse fly, but also indicate that infections established by slender cells can still result in efficient completion of the passage through the tsetse fly.

## In the tsetse midgut, dividing slender bloodstream stage parasites activate the PAD1 pathway and differentiate to the procyclic insect stage without arresting the cell cycle

To determine how pleomorphic slender trypanosomes manage to establish infections, we observed the early events following trypanosome ingestion by tsetse flies (*Figure 4—video 1*). The canonical version of events is that ingested stumpy (i.e. PAD1-positive) cells reactivate the cell cycle, begin to express the EP procyclin protein on their cell surface, and differentiate to the procyclic life cycle stage (*Dean et al., 2009*; *Matthews and Gull, 1994*; *Mowatt and Clayton, 1987*; *Richardson et al., 1988*; *Roditi et al., 1989*; *Ziegelbauer and Overath, 1990*). We infected tsetse flies with pleomorphic trypanosomes which not only contained the stumpy-specific GFP:PAD1$^{UTR}$ marker, but also encoded an EP1:YFP fusion (*Figure 4*; *Engstler and Boshart, 2004*). In this way, the onset of stumpy development was observable as GFP fluorescence in the nucleus, and further

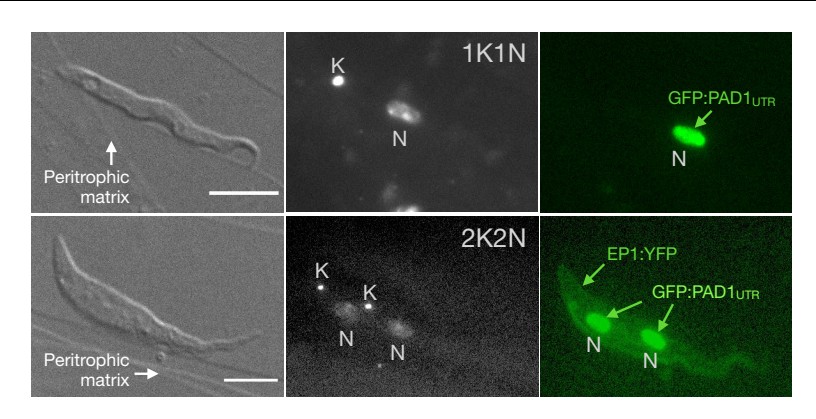

**Figure 4.** Exemplary images of procyclic trypanosomes in the explanted tsetse midgut 24 hr post infection with slender cells. Morphology (DIC panels, left), cell cycle status (DAPI label, middle panels) and expression of fluorescent reporters (right) were scored. Note that the upper panels show a cell with procyclic morphology that is nonetheless EP1:YFP negative, indicating that the EP1 signal underestimates the total numbers of procyclic cells in the population. Scale bar: 5 μm.

The online version of this article includes the following video for figure 4:

**Figure 4—video 1.** After uptake by the tsetse fly, slender trypanosomes promptly activate the PAD1 pathway, while the continuously dividing.

https://elifesciences.org/articles/66028#fig4video1

differentiation to the procyclic life cycle stage as YFP fluorescence on the parasite cell surface. In addition, the cell cycle status (K/N counts, see *Figure 1A*), morphology, and the characteristic motile behavior of the trypanosomes were also assessed as criteria of developmental progress. In total, 114 tsetse flies (57 male and 57 female) were dissected after at least six independent infections with either 12,000 slender or stumpy parasites each. These high initial parasite numbers allowed the microscopic analysis of individual living slender (n = 1845) and stumpy trypanosomes (n = 1237) within the complex microenvironment of midgut explants (*Schuster et al., 2017*). As early as 2 hr post-infection with slender trypanosomes, a few (0.8%) 2K1N dividing trypanosomes with a nuclear PAD1 signal could be observed (*Figure 5A*, blue). After 8 hr however, half (38.3+6.8+5.3=50.4%) of all trypanosomes in the explants were PAD1-positive (*Figure 5A*; *Figure 5—figure supplement 1* shows summed cell cycle category values for PAD1-positive cells). After 24 hr, 84.3% (56.3+15.0 +13.0) of the parasites expressed PAD1. Of these, 9.8% had already initiated developmental progression to the procyclic insect stage, as evidenced by EP1:YFP fluorescence on their cell surface (*Figure 6*, blue). At 48 hr post-infection with slender trypanosomes, virtually the entire trypanosome population (91.8%) expressed PAD1, and almost one fifth (19.1%) of cells were EP1-positive (*Figure 6*). To examine cell cycle progression, we counted the number of 1K1N, 2K1N, and 2K2N cells in the PAD1-positive and PAD1-negative slender cell populations. Remarkably, 15 hr post-infection, the majority of all replicating (i.e. 2K1N + 2K2N) cells were PAD1-positive (*Figure 5B*; *Figure 5—figure supplement 1*). No indication of a transient cell cycle arrest or intermittent impairment of cell cycle progression was observed. Over the duration of the experiment, PAD1-negative cells gradually decreased in numbers, while PAD1-positive slender cells were increasingly observed at all cell cycle stages (*Figure 5B* blue vs. grey; *Figure 4—video 1C*). After 2 days, more than 90% of dividing trypanosomes were PAD1-positive. Thus, the PAD1 pathway was triggered in replicating slender trypanosomes upon ingestion by the fly, without prior or subsequent cell cycle arrest.

In order to directly compare the kinetics of slender-to-procyclic differentiation with that of stumpy stage trypanosomes, we fed flies with SIF-induced, PAD1-positive stumpy trypanosomes (*Figure 5A*, red). These cells remained as 1K1N cells in cell cycle arrest for the first day, then differentiated to procyclic cells and re-entered the cell cycle after 2 days. Four hours after uptake by the tsetse fly, stumpy trypanosomes started expressing EP1:YFP (*Figure 6*, red). The fluorescent reporter was visible on 16.2% of stumpy cells after 10 hr, showing that EP expression was initiated before release of

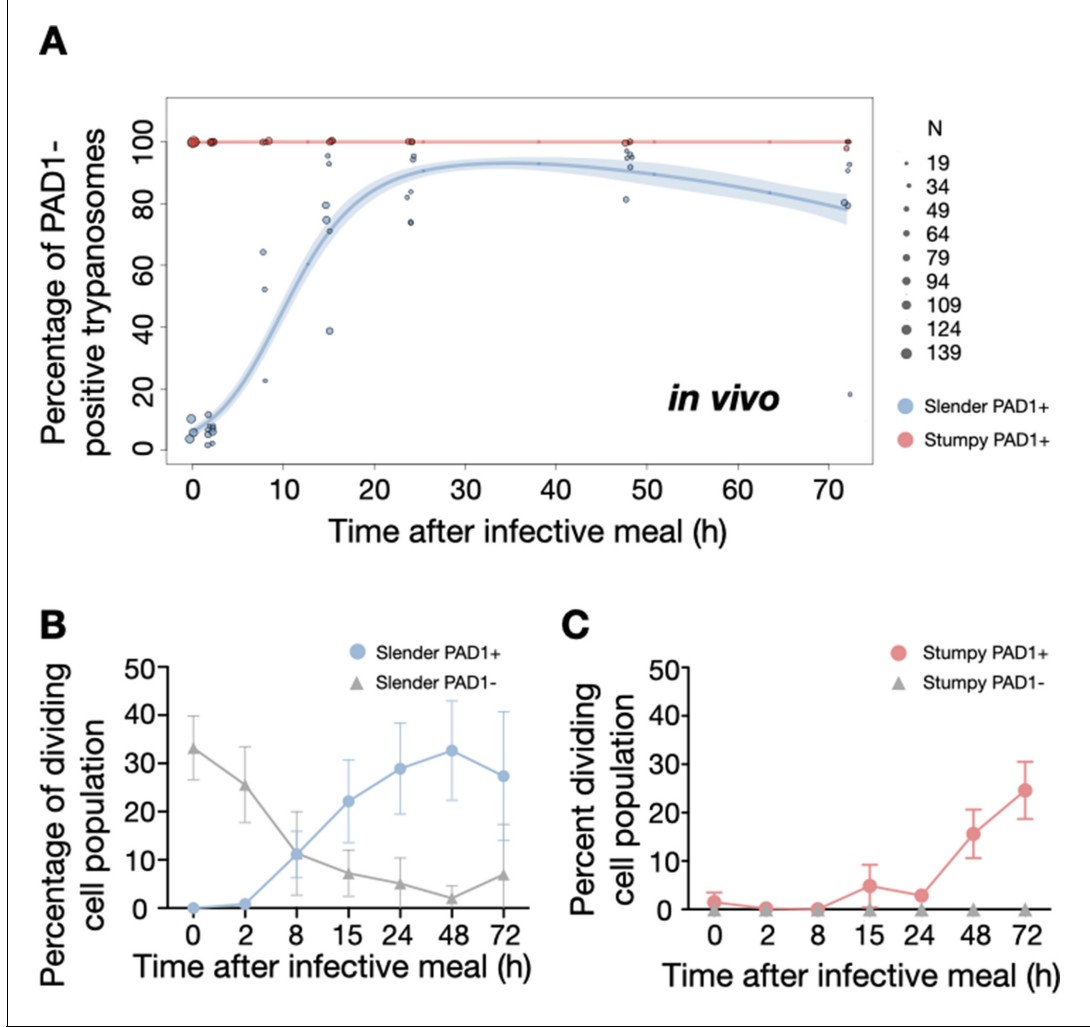

**Figure 5.** Slender trypanosomes activate the PAD1 pathway upon uptake by the tsetse fly. Tsetse flies were infected with either slender (3.6% PAD1-positive) or stumpy (100% PAD1-positive) trypanosomes. 72 (slender) or 42 (stumpy) flies were dissected (equal sex ratios) at different timepoints after infection (for each time point, one hour was given to either slender or stumpy infected flies for dissection and cell analysis). Experiments were done at least three times. Living trypanosomes (>100 cells per time point) were microscopically analysed in the explanted tsetse midguts and scored for the expression of the fluorescent stumpy reporter GFP:PAD1UTR in the nucleus. Stumpy cells (n=1237) are red, and slender cells (n=1845) are blue. (A) Percentages of PAD1-positive slender and stumpy cells over time after uptake by the tsetse fly. Points indicate the individual experiments for either slender (blue) or stumpy (red). Point sizes correspond to the total number of cells counted per experiment. These data were fed into a *point estimate model* and are shown as solid lines, indicating the predicted percentage of PAD1-positive cells, based on time vs. cell type. Transparent colours indicate the associated 95% confidence bands. The difference between slender and stumpy cells over time is strongly significant (p<0.001). (B, C) Slender and stumpy trypanosomes scored as PAD1-positive or -negative were also stained with DAPI, and the cell cycle position determined based on the configuration of kinetoplast (K) to nucleus (N) at the timepoints indicated. The dividing slender population (B) and dividing stumpy population (C) are shown. As seen, the percentage of PAD1-positive slender cells steadily increased (B, blue) while the percentage of PAD1-negative cells steadily decreased (B, grey). This shows that slender cells can seamlessly turn on the PAD1 pathway, within a continuously dividing population. Stumpy cells did not show a normal cell cycle profile until 48 hr after tsetse uptake (C, red), as the cells differentiated to the procyclic stage. They did however remain PAD1-positive even as dividing parasites at 72 hr. Data are shown as mean +/- SD. Points without SD were the result of two measurements at those timepoints.

The online version of this article includes the following figure supplement(s) for figure 5:

**Figure supplement 1.** Slender populations exhibit continuous division while turning on the PAD1 pathway.

**Figure supplement 2.** Slender cells survive in the tsetse midgut at early timepoints after infection.

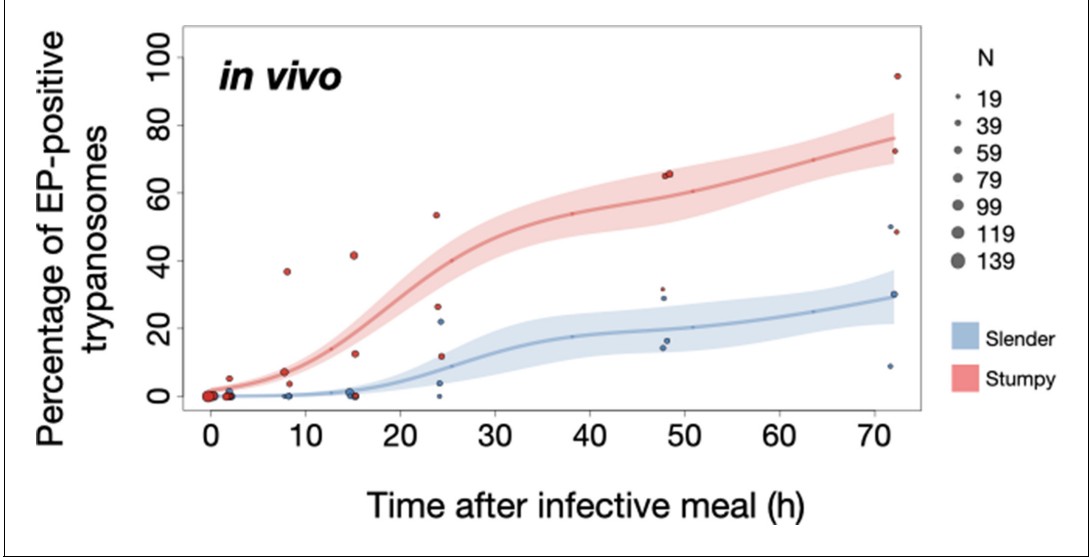

**Figure 6.** Slender trypanosomes show delayed expression of EP compared to stumpy trypanosomes, while directly differentiating to the procyclic life cycle stage in the tsetse fly. Tsetse flies were infected with either slender (3.6% PAD1-positive) or stumpy (100% PAD1-positive) trypanosomes. 72 (slender) or 42 (stumpy) flies were dissected (equal sex ratios) at different timepoints after infection (for each time point, one hour was given to either slender or stumpy infected flies for dissection and cell analysis). Experiments were done at least three times. Living trypanosomes (>100 cells per time point) were microscopically analysed in explanted tsetse midguts and scored for the procyclic insect stage reporter EP1:YFP on the cell surface. Stumpy cells (n=1237) are shown in red and slender cells (n=1845) in blue. Points indicate the individual cell counts for either slender (blue) or stumpy (red). Point sizes correspond to the total number of cells counted per experiment. These data were fed into a *point estimate model* (see Materials and methods) and are shown as solid lines, indicating the predicted percentage of EP-positive cells, based on time vs. cell type. Shading above and below the lines indicates the associated 95% confidence bands. The difference between slender and stumpy cells over time was strongly significant (p<0.001).

cell cycle arrest (*Figure 5C*, red; *Figure 6*, red). Uncoupling of EP surface expression from the commitment to differentiation has been reported before (*Engstler and Boshart, 2004*).

EP1:YFP expression in slender parasites lagged 12 hr behind stumpy cells, only becoming widespread after 24 hr (*Figure 6*). Thus, the onset of EP1 expression was shifted, but the kinetics of differentiation were comparable in slender and stumpy parasites, and activation of the PAD1 pathway also preceded developmental progression in slender cells. This strongly suggests that expression of PAD1 is essential for differentiation to the procyclic stage, while cell cycle arrest is not.

Of note, EP1 expression did not directly correlate with acquisition of procyclic morphology. At 24 hr, 9.8% of slender cells were EP1-positive (*Figure 6*), but the EP1-negative cells frequently exhibited procyclic morphology (*Figure 4*, upper panels). GPEET is another procyclic surface protein that is expressed early in the transition from bloodstream stage to procyclic stage cells in the tsetse midgut, before being replaced by EP (*Vassella et al., 2000*). Whether these early and morphologically procyclic cells expressed GPEET was not checked, and remains a target of future work. An example of a dividing (2K2N), PAD1-positive, EP1-positive cell is also shown (*Figure 4*, lower panels; *Figure 4—video 1*). This strongly suggests that a seamless developmental stage transition from the slender bloodstream stage to the procyclic insect stage took place, which was accompanied by the typical re-organisation of the cytoskeleton and the concomitant switch of swimming styles (*Rotureau et al., 2011*; *Heddergott et al., 2012*; *Schuster et al., 2017*).

## Pleomorphic slender bloodstream stage trypanosomes can seamlessly differentiate to the procyclic insect stage without preceding cell cycle arrest in vitro

The factor(s) or condition(s) that trigger differentiation of bloodstream stage trypanosomes to the procyclic insect stage in the tsetse midgut are still ill-defined. In the laboratory, differentiation to the procyclic insect stage is routinely induced by the addition of *cis*-aconitate, removal of glucose, and a

temperature drop from 37°C to 27°C (*Brun et al., 1981*; *Czichos et al., 1986*; *Engstler and Boshart, 2004*; *Qiu et al., 2018*; *Ziegelbauer et al., 1990*).

We used this protocol to further investigate the developmental potential of cultivated pleomorphic slender bloodstream stage in vitro using the same cell lines and analysis as above (*Figure 7A*). Slender trypanosomes activated the PAD1 pathway rapidly after receiving the trigger, with 9.8% of all parasites being PAD1-positive within 2 hr, and 83.2% after 10 hr. PAD1 expression peaked after one day (98.3%), and declined thereafter (*Figure 7A*; *Figure 7—figure supplement 1*). Shortly after PAD1 reporter expression, EP1 appeared on the cell surface of 19.6% of all parasites within 8 hr, increasing to 98.3% after 3 days (*Figure 8*). PAD1 and EP protein appearance on the cell surface was monitored throughout the timecourse using immunofluorescence (*Figure 7—figure supplement 2*). Throughout the timecourse, PAD1-positive 2K1N and 2K2N cells were continually observed, demonstrating that the PAD1-positive slender parasites did not arrest in the cell cycle, and continued dividing throughout in vitro differentiation to the procyclic stage (*Figure 7B*, blue). The acquisition of procyclic identity was further confirmed by measuring the distance between the nucleus and cell posterior, with lengths matching those of procyclic cells being reached within 72 hr (*Figure 7—figure supplement 3*). After 3 days of differentiation treatment in vitro, slender trypanosomes had established a proliferating procyclic parasite population, which was further confirmed using morphological analysis (*Figure 7—figure supplement 3*).

By comparison, stumpy parasites (*Figure 7A*, red) responded to in vitro *cis*-aconitate treatment with rapid expression of the EP1:YFP marker, with 28.6% of all cells being positive within 2 hr (*Figure 8*). After 1 day, EP1 was present on almost all (96.7%) stumpy trypanosomes. The cell cycle analysis revealed that these parasites were not dividing, however (*Figure 7C*). The first cells re-entered the cell cycle only after 15 hr, and a normal procyclic cell cycle profile was not reached until day 3. Morphological analysis using the distance between the nucleus and the posterior of the cell confirmed acquisition of procyclic identity within 72 hr (*Figure 7—figure supplement 3*). Thus, the in vitro differentiation supported the in vivo observations, demonstrating that pleomorphic slender trypanosomes are able to directly differentiate to the procyclic stage without becoming cell cycle-arrested stumpy cells. The surface expression of EP1 is also of note: it has been shown that in slender bloodstream parasites, ectopically expressed EP1 does not enter the cell surface, but is retained in endosomes and the flagellar pocket (*Engstler and Boshart, 2004*). Hence, as in stumpy trypanosomes, lifting of the cell surface access block for EP1 in the slender trypanosome correlates with activation of the PAD1 pathway. To summarise, the overall developmental capacity of the two life cycle stages - slender and stumpy cells - is comparable both in vitro and in vivo.

## Discussion

Our observations suggest a revised view of the life cycle of African trypanosomes (*Figure 9*). We have shown that one trypanosome suffices to produce robust infections of the tsetse vector, and that the stumpy stage is not essential for tsetse transmission. Slender parasites can complete the complex life cycle in the fly with comparable overall success rates and kinetics as the stumpy stage. Interestingly, the stumpy stage appears more able to establish initial infections in the fly midgut (*Figure 2A*, column 5, MG), while slender-derived parasites appear to produce salivary gland infections slightly more efficiently (higher amount of salivary gland infections compared to midgut infections, TI) than stumpy-derived counterparts (*Figure 2A*, column 8, TI). At first sight, this discrepancy may be related to a greater resistance of the stumpy stage to the digestive environment in the fly's gut, as has been suggested (*Matetovici et al., 2019*; *Nolan et al., 2000*). This, however, is not supported by our data. We have not observed cell death of monomorphic or pleomorphic slender cells in infected tsetse midguts (*Figure 5—figure supplement 1*). And even if so, why then should slender-derived cells perform better in the second part of the life cycle? As there will not be a difference between slender- and stumpy-derived procyclic cells, the difference observed must be based on the behaviour of bloodstream parasites in the midgut. It is tempting to speculate that one decisive factor could be trypanosome motility. Slender trypanosomes exhibit significantly higher motility compared to stumpy trypanosomes (*Bargul et al., 2016*). Thus, the mean square displacement in the midgut will be much larger for slender parasites. While stumpy trypanosomes probably never reach the 'midgut exit' before differentiation to the insect stage, slender trypanosomes could already be located close to the proventriculus before starting to differentiate to the procyclic stage. Thus,

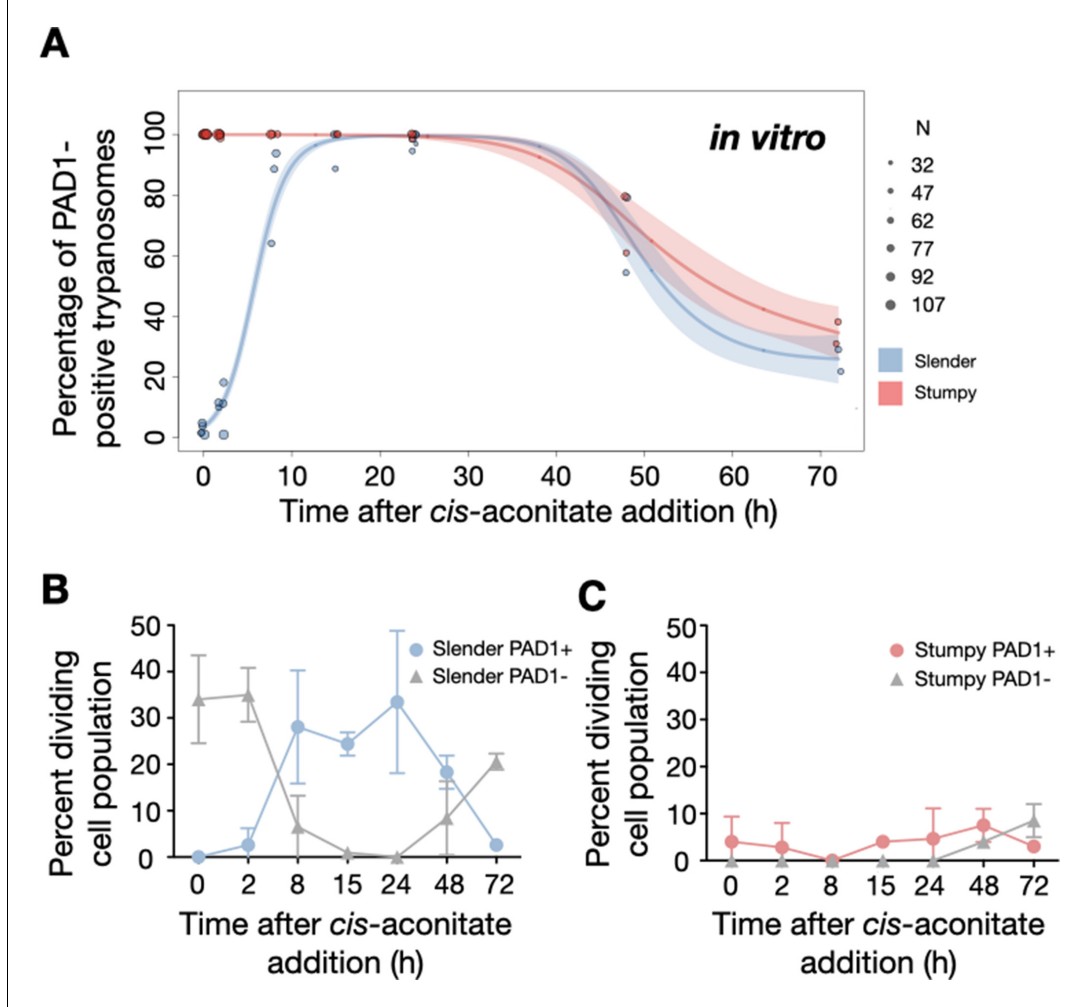

**Figure 7.** Slender trypanosomes activate the PAD1 pathway in vitro with a continuously dividing population. Cultured slender or stumpy trypanosomes were differentiated in vitro by the addition of *cis*-aconitate and a temperature reduction to 27˚C. At the times indicated, trypanosomes were analysed for the expression of the fluorescent stumpy reporter GFP:PAD1$^{UTR}$, as in *Figure 5*. Slender cells (n=1653) are shown in blue and stumpy cells (n=1798) in red. Data were compiled from five independent experiments, with each time point being analysed in at least two separate experiments. (A) Percentages of PAD1-positive slender and stumpy cells over time. Points indicate the individual experiments for either slender (blue) or stumpy (red) trypanosomes. Point sizes correspond to the total number of cells counted per experiment. These data were fed into a *point estimate model,* and are shown as solid lines, indicating the predicted percentage of PAD1-positive cells, based on time vs. cell type. Transparent colours indicate the associated 95% confidence bands. The difference between slender and stumpy cells over time was strongly significant (p<0.001). (B, C) Slender and stumpy trypanosomes scored as PAD1-positive or -negative were also stained with DAPI, and the cell cycle position determined based on the configuration of kinetoplast (K) to nucleus (N) at the timepoints indicated. The dividing slender population (B) and dividing stumpy population (C) are shown. As seen, the percentage of PAD1-positive slender cells steadily increased (B, blue) and the percentage of PAD1-negative cells steadily decreased (B, grey). This shows that slender cells can turn on the PAD1 pathway without apparent cell cycle arrest. Although a small portion of the stumpy population was observed to to divide throughout the time points (C, red), cells did not return to a normal cell cycle profile until 48 hr after the addition of *cis*-aconitate. As the cells became more procyclic, they began to lose their PAD1 signal and an increase in PAD1-negative dividing cells was seen (B, C, grey). Data are shown as mean +/- SD. Points without SD were the result of two measurements at those timepoints.

The online version of this article includes the following figure supplement(s) for figure 7:

**Figure supplement 1.** Slender trypanosomes activate the PAD1 pathway in vitro with a continuously dividing cell population.

**Figure supplement 2.** Slender cells express PAD1 and EP on their surface after the addition of cis-aconitate.

**Figure supplement 3.** Slender trypanosomes exhibit procyclic morphology on the same time scale as stumpy trypanosomes when differentiating.

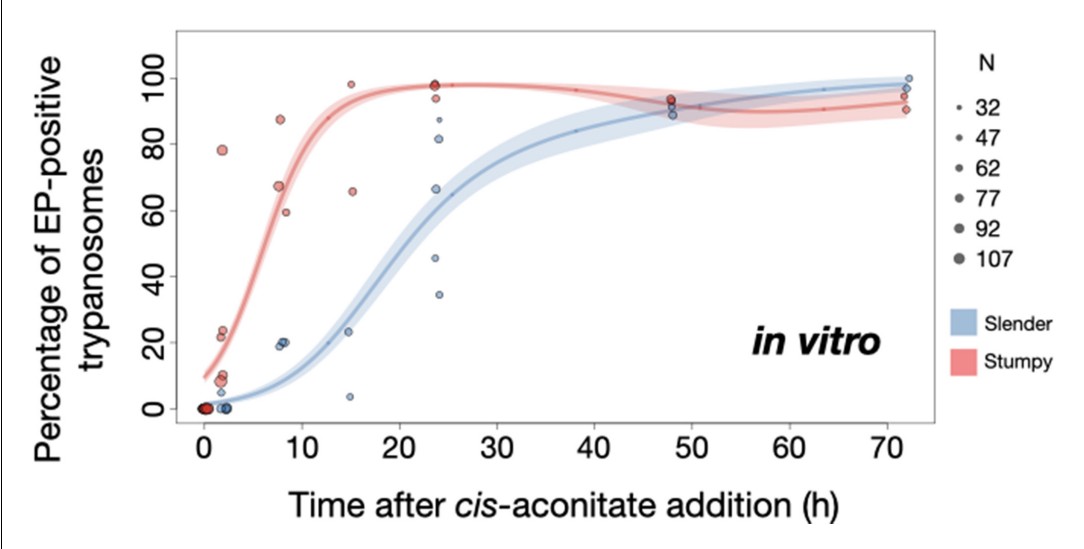

**Figure 8.** Slender trypanosomes show delayed expression of EP while they differentiate to the procyclic life cycle stage in vitro. Cultured slender or stumpy trypanosomes were differentiated in vitro by the addition of *cis*-aconitate and a temperature reduction to 27°C. At the times indicated, trypanosomes were analysed for the expression of the procyclic fluorescent reporter EP1:YFP, as in *Figure 6*. Stumpy cells (n=1798) are shown in red and slender cells (n=1653) in blue. Points indicate the individual experiments for either slender or stumpy. Point sizes correspond to the total number of cells counted per experiment. These data were fed into a *point estimate model* and are shown as solid lines, indicating the predicted percentage of EP-positive cells, based on time vs. cell type. Shading above and below the lines indicates the associated 95% confidence bands. The difference between slender and stumpy cells over time was strongly significant (p<0.001).

passage through the proventriculus could occur immediately, and the slender-derived trypanosomes could rapidly progress to the mesocyclic stage. This faster mesocyclic progression would result in a less-pronounced infection of the midgut, and a higher TI-value for the slender-derived trypanosomes. While the above hypothesis is consistent with our data, experimental proof would be extremely challenging to obtain. The recent demonstration that glucose levels are a developmental trigger in addition to the well-characterised ones of cold shock and *cis*-aconitate adds another layer of complexity to the early events during tsetse infection (*Qiu et al., 2018*). It is also of note that a recent review has suggested that stumpy trypanosomes may have evolved as a stress response, with procyclic pathways already activated in order to increase chances of survival in the fly (*Quintana et al., 2021*).

The dogma that cell cycle-arrested stumpy cells are the only trypanosomes that infect the tsetse fly has never been experimentally challenged, although there are quite a number of reports that point against an exclusive role for stumpy parasites in the life cycle. Koch's detailed report on the activities of the German sleeping sickness commission sent to East Africa in 1906/7 states that the trypanosome numbers in the blood of human sleeping sickness infections was always very low (*Koch, 1909*). From his data, we have calculated an average blood parasitaemia between 10 and 100 trypanosomes cells/ml (see Appendix 1). This means that two or fewer trypanosomes would be present in an average tsetse bloodmeal, again highlighting the rarity of a tsetse taking up a stumpy cell. In 1930, Duke discussed the evidence for the essential status of the stumpy stage for tsetse transmission, and his data did not support it (*Duke, 1930*). Further, Baker and Robertson in 1957 compared the infection capability of *T. rhodesiense* and *T. brucei* using guinea pig feeding (*Baker and Robertson, 1957*). They concluded: 'Neither the morphology nor the intensity of the parasitaemia in the infecting mammal was obviously related to the subsequent infection rates in the tsetse-flies'. In 1990, Bass and Wang suggested that in fact the stumpy stage may be dispensable for development to the insect stage (*Bass and Wang, 1991*). The experiments, however, were in part inconclusive, mainly because a molecular marker for the stumpy stage was missing. The discovery of SIF in the 1990s and the realisation that quorum sensing underpinned the differentiation to the stumpy stage led to an assumption that the slender stage had no role to play in the transmission event. Subsequent research has been focused on the details of stumpy formation, while the

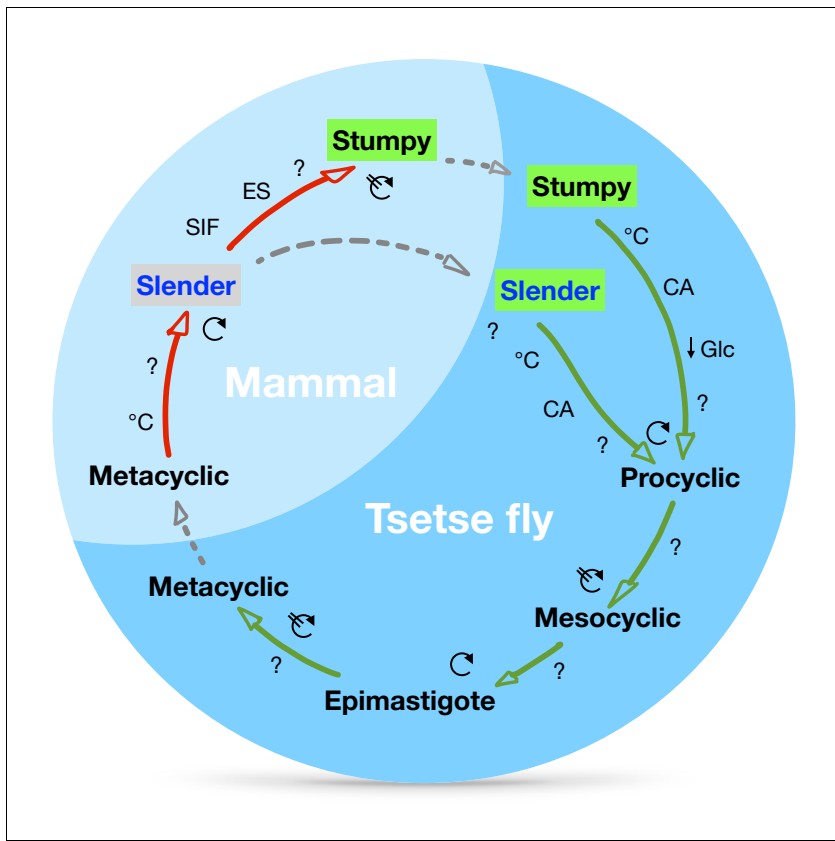

**Figure 9.** A revised life cycle for the parasite *Trypanosoma brucei*. Cell-cycle-arrested ($G_0$) metacyclic trypanosomes are injected by the tsetse fly into the mammalian host's skin. There, the parasites re-enter the cell cycle, and proliferate as ( ) slender forms in the blood, while disseminating into the interstitium and various tissues, including fat and brain. At least two triggers (SIF or ES) launch the PAD1-dependent differentiation pathway (light green boxes) to the cell cycle-arrested ( ) stumpy bloodstream stage. Stumpy trypanosomes can establish a fly infection when taken up with the bloodmeal of a tsetse. The work described here reveals that proliferating slender stage trypanosomes are equally effective for tsetse transmission, that a single parasite suffices, and that the population can continuously divide while differentiating to the procyclic insect stage. The triggers that initiate further developmental transitions are temperature (˚C), *cis*-aconitate (CA) and glucose deprivation (↓Glc).

developmental role of the stumpy cell has not undergone further examination. The above publications all relate to what is nowadays referred to as the transmission paradox, the persistence and circulation of trypanosomiasis in a population even when parasitaemia levels in individuals are low or close to elimination (*Capewell et al., 2019*). When parasitaemia is low, stumpy trypanosomes are characteristically absent, making the probability of being ingested by a tsetse fly (which on average ingests 20 µl of blood) extremely low. Yet trypanosomiasis persists, even when statistically it should be eliminated. Solutions to the paradox have long been hypothesised and variously include flawed diagnostic testing, asymptomatic cases, and animal reservoirs (*Alvar et al., 2020*). Recent work using theoretical modelling suggests that for *T. gambiense*, trypanosomes residing in the skin of humans could solve the problem (*Capewell et al., 2019*). However, there are currently no data available on the number of trypanosomes located in asymptomatic human skin, nor have the kinetics of fly uptake of skin-localised trypanosomes been explored. Also, different tsetse-transmitted trypanosome species reveal rather distinct distributions in the host, such as *Trypanosoma congolense* preferentially residing in small blood vessels (*Banks, 1978*). Thus, while trypanosomes in the skin may be important for the persistence of the parasites, their existence alone does not automatically solve the transmission paradox. As tsetse are blood pool feeders, it actually does not matter if the trypanosomes reside in the skin, fat tissue, or blood.

Furthermore, stumpy cells will inevitably run into an age-related problem. They are not replicative, and their lifetime is limited to roughly 3 days (*Turner et al., 1995*). In the fly, re-entry into the

cell cycle is by no means immediate, but takes at least one day. Following induction of cell cycle arrest, the stumpy cells would need to be taken up by the fly within 1 day. Thus, only a subset of rather young stumpy cells would prove successful in the midgut. It is important to note that this is not the case in our experiments, as only freshly differentiated stumpy cells were used for tsetse infection. Thus, our experiments in fact overestimate the success rate of stumpy stage trypanosomes.

It should be stressed that our data provide a possible solution to the transmission paradox without challenging any of the extensive published work on stumpy trypanosomes. We have shown that slender and stumpy trypanosomes are equally competent for fly passage. The PAD1 pathway has an essential role in preparing both bloodstream stages for differentiation to the procyclic cell stage. For successful passage through the tsetse fly, however, the stumpy stage is not uniquely required. Along similar lines, it is worth noting that *Trypanosoma congolense*, the principal causative agent of the cattle plague nagana, infects tsetse flies without manifesting a cell cycle-arrested stumpy stage (*Rotureau and Van Den Abbeele, 2013*). Thus, the essential biological function of the stumpy life cycle stage in *T. brucei* may not be transmission, but rather quorum sensing (SIF)-dependent control of population size in the host. This pathway can be triggered in other ways, and even at low levels of parasitaemia, for example by VSG expression site attenuation (ES)(*Zimmermann et al., 2017*). The capacity for inducing cell cycle arrest at the single cell level might actually have been important for the evolution of antigenic variation. As not all trypanosome species develop a stumpy life cycle stage (*Rotureau and Van Den Abbeele, 2013*), density-dependent differentiation at the population level may well be a later innovation in evolution, and specific to the *T. brucei* group. In conclusion, our work evidences a high degree of plasticity in the life cycle of an important parasite. It shows that the trypanosome life cycle is not rigid, and we have proposed a revised and more flexible view of the trypanosome life cycle that may help solve a longstanding puzzle in parasitology.

# Materials and methods

**Key resources table**

| Reagent type (species) or resource | Designation | Source or reference | Identifiers | Additional information |
|---|---|---|---|---|
| Cell line (*Trypanosoma brucei brucei*) | 'SIF' cell line, EATRO 1125 Antat1.1 (GFP:PAD1$^{UTR}$) | The GFP:PAD1$^{UTR}$ reporter was verified in: *Zimmermann et al., 2017* | DOI:10.1371/journal.ppat.1006324 | Transfected with: p4231 (containing an NLS-GFP reporter fused to the 3' UTR of the PAD1 gene, which inserts into the β-tubulin locus; provided by Mark Carrington) |
| Cell line (*Trypanosoma brucei brucei*) | 'ES' cell line, Antat 1.1 13–90 (GFP:PAD1$^{UTR}$) | This cell line was verified in: *Zimmermann et al., 2017* | DOI:10.1371/journal.ppat.1006324 | Transfected with: pLew13 (Addgene plasmid 24007, G. Cross); pLew90 (Addgene plasmid 24008, G. Cross); p4231 (containing an NLS-GFP reporter fused to the 3' UTR of the PAD1 gene, which inserts into the β-tubulin locus; provided by Mark Carrington) |
| Cell line (*Trypanosoma brucei brucei*) | 'SIF' cell line, EATRO 1125 Antat1.1 (GFP:PAD1$^{UTR}$) with an EP1:YFP fusion protein | The GFP:PAD1$^{UTR}$ reporter was verified in: *Zimmermann et al., 2017* The EP1:YFP reporter was verified in: *Engstler and Boshart, 2004* | DOI:10.1371/journal.ppat.1006324 DOI: 10.1101/gad.323404 | Transfected with: p4231 (containing an NLS-GFP reporter fused to the 3' UTR of the PAD1 gene, which inserts into the β-tubulin locus; provided by Mark Carrington); pGaprone(ble)_EPGFG_rev GPY PARPYFP |

*Continued on next page*

*Continued*

| Reagent type (species) or resource | Designation | Source or reference | Identifiers | Additional information |
|---|---|---|---|---|
| Cell line (*Trypanosoma brucei brucei*) | Monomorphic cell line, *T. brucei* 427 MITat 1.2 13–90 | This cell line was verified in: *Wirtz et al., 1999* | DOI: 10.1016/s0166-6851(99)00002-x | |
| Antibody | Anti-PAD1 (*Trypanosoma brucei brucei*) (Rabbit Polyclonal) | This antibody was generated and verified in: *Dean et al., 2009* | DOI: 10.1038/nature07997 | IF (1:100) |
| Antibody | Anti-EP1 (*Trypanosoma brucei brucei*) (Mouse monoclonal) | CEDARLANE | Product code: CLP001A | IF (1:500) |
| Chemical compound, drug | Methylcellulose | Sigma Aldrich | Product code: 94378 | Methocel A4M |
| Software, algorithm | R | R core team | | Package 'mgcv' |
| Software, algorithm | ImageJ | NIH | DOI: 10.1038/nmeth.2089 | |

## Trypanosome culture

Pleomorphic *Trypanosoma brucei brucei* strain EATRO 1125 (serodome AnTat1.1)(*Le Ray et al., 1977*) bloodstream stages were grown in HMI-9 medium (*Hirumi and Hirumi, 1989*), supplemented with 10% (v/v) foetal bovine serum and 1.1% (w/v) methylcellulose (Sigma 94378, Munich, Germany) (*Vassella et al., 2001*) at 37°C and 5% $CO_2$. Slender stage parasites were maintained at a maximum cell density of $5x10^5$ cells/ml. For cell density-triggered differentiation to the stumpy stage, cultures seeded at $5x10^5$ cells/ml were cultivated for 48 hr without dilution. Within this period, the stumpy induction factor (SIF) accumulated and caused developmental transition of slender to stumpy trypanosomes. For expression site attenuated differentiation (ES stumpy cells) to the stumpy stage (ES stumpy), cells with a construct which overexpresses VSG121 upon tetracycline induction were grown to $5x10^5$ cells/ml and incubated with 1 µg/ml tetracycline for 56 hr (*Zimmermann et al., 2017*). Pleomorphic parasites were harvested from the viscous medium by 1:4 dilution in trypanosome dilution buffer (TDB; 5 mM KCl, 80 mM NaCl, 1 mM $MgSO_4$, 20 mM $Na_2HPO_4$, 2 mM $NaH_2PO_4$, 20 mM glucose, pH 7.6), followed by filtration (MN 615 ¼, Macherey-Nagel, Dueren, Germany) and centrifugation (1400x*g*, 10 min, 37°C)(*Zimmermann et al., 2017*). For infections with very low trypanosomes per bloodmeal (0.2, 1), cells were pipetted directly from their culture medium and placed in blood. Monomorphic *T. brucei* 427 MITat 1.2 13–90 bloodstream stage (*Wirtz et al., 1999*) were grown in HMI-9 medium (*Hirumi and Hirumi, 1989*), supplemented with 10% (v/v) foetal bovine serum at 37°C and 5% $CO_2$.

For in vitro differentiation to the procyclic insect stage, bloodstream stage trypanosomes were pooled to a cell density of $2x10^6$ cells/ml in DTM medium with 15% fetal bovine serum immediately before use (*Overath et al., 1986*). *Cis*-aconitate was added to a final concentration of 6 mM (*Brun et al., 1981*; *Overath et al., 1986*) and temperature was adjusted to 27°C. Procyclic parasites were grown in SDM79 medium (*Brun and Schönenberger, 1979*), supplemented with 10% (v/v) foetal bovine serum (*Hirumi and Hirumi, 1989*) and 20 mM glycerol (*Schuster et al., 2017*; *Vassella et al., 2000*).

## Genetic manipulation of trypanosomes

Transfection of pleomorphic trypanosomes was done as previously described (*Zimmermann et al., 2017*), using an AMAXA Nucleofector II (Lonza, Basel, Switzerland). Transgenic trypanosome clones were selected by limiting dilution in the presence of the appropriate antibiotic. The GFP:PAD1UTR reporter construct (*Zimmermann et al., 2017*) was used to transfect AnTat1.1 trypanosomes to yield the cell line 'SIF'. The trypanosome 'ES' line was described previously (*Zimmermann et al., 2017*). It contains the reporter GFP:PAD1UTR construct and an ectopic copy of VSG gene MITat 1.6 under the

control of a tetracycline-inducible T7-expression system. The EP1:YFP construct was integrated into the EP1-procyclin locus as described previously (*Engstler and Boshart, 2004*).

## Immunofluorescence

Cells were harvested as stated above, concentration was measured using a Neubauer chamber, and $10^6$ cells per coverslip were taken. The cells were transferred to a 1.5 ml tube, washed twice with 1 ml of phosphate buffered saline (PBS), resuspended in 500 µl of PBS, and fixed by addition of para-formaldehyde to a final concentration of 4% at room temperature (RT) for 20 min. The cells were pelleted by centrifugation (750 $xg$, RT, 10 min), the supernatant removed, and the cell pellet resuspended in PBS and transferred to poly-L-lysine-coated coverslips in a 24-well plate. Cells were attached to coverslips by centrifugation (750 $xg$, RT, 4 min). Cells were either permeabilised with 0.25% TritonX-100 in PBS (RT, 5 min) and subsequently washed twice with PBS or not permeabilised, so as to allow only surface labelling. Cells were then blocked with 3% BSA in PBS (RT, 30 min), followed by incubation with the primary antibody (1:100 rabbit anti-PAD1; 1:500 IgG1 mouse anti-*Trypanosoma brucei* procyclin, Ascites, Clone TBRP1/247, CEDARLANE, Ontario, Canada) followed by secondary antibodies (Alexa488- and Alexa 594-conjugated anti-rabbit and anti-mouse, 1:100, ThermoFisher Scientific, Massachusetts, USA) diluted in PBS (1 hr, RT for each), with three PBS wash steps after each incubation. After the final wash, coverslips were rinsed with ddH$_2$O, excess fluid removed by wicking, and mounted on glass slides using antifade mounting media with DAPI (Vectashield, California, USA).

## Tsetse maintenance

The tsetse fly colony (*Glossina morsitans morsitans*) was maintained at 27°C and 70% humidity. Flies were kept in Roubaud cages and fed three times a week through a silicone membrane, with pre-warmed, defibrinated, sterile sheep blood (Acila, Moerfelden, Germany).

## Fly infection and dissection

Teneral flies were infected 1–3 days post-eclosion during their first meal. It is known that teneral flies (flies that are newly hatched and unfed) are more susceptible to midgut infections compared to older flies, and it is an accepted practice in the field to use teneral flies for infections. While all of our infections were done during the flies' first bloodmeal, it is of note that 1–3 days is rather old for teneral flies (*Walshe et al., 2011*; *Wijers, 1958*). Depending on the experiment, trypanosomes were diluted in either pre-warmed TDB or sheep blood. For infections with low parasite numbers (0.2 and 1 cell/bloodmeal; *Figure 2A*), the cell density of either stumpy or slender trypanosomes was calculated, and the dilutions made directly in blood. Thus, the parasites were directly taken from culture and added to blood, thereby completely omitting any filtration step.

The infective meals were supplemented with 60 mM *N*-acetylglucosamine (*Peacock et al., 2006*). For infection with 2400 monomorphic parasites per bloodmeal, cells were additionally treated for 48 hr with 12.5 mM glutathione (*MacLeod et al., 2007*) and 100 µM 8-pCPT-cAMP (cAMP) (*Vassella et al., 1997*).

Tsetse infection status was analysed between 35 and 40 days post-infection. Flies were euthanised with chloroform and dissected in PBS. Intact tsetse alimentary tracts were explanted and analysed microscopically, as described previously (*Schuster et al., 2017*). For the analysis of early trypanosome differentiation in vivo, slender or stumpy trypanosomes at a concentration of $6\times10^5$ cells/ml were resuspended in TDB to the required final concentration and fed to flies. The numbers of flies used and the number of independent experiments carried out are indicated in the figure legends. Results are presented as sample means.

## Fluorescence microscopy and video acquisition

Live trypanosome imaging was performed with a fully automated DMI6000B widefield fluorescence microscope (Leica microsystems, Mannheim, Germany), equipped with a DFC365FX camera (pixel size 6.45 µm) and a 100x oil objective (NA 1.4). For high–speed imaging, the microscope was additionally equipped with a pco.edge sCMOS camera (PCO, Kelheim, Germany; pixel size 6.5 µm). Fluorescence video acquisition was performed at frame rates of 250 fps. For visualisation of parasite cell cycle and morphology, slender and stumpy trypanosomes were harvested and incubated with 1 mM

AMCA-sulfo-NHS (Thermo Fisher Scientific, Erlangen, Germany) for 10 min on ice. Cells were chemically fixed in 4% (w/v) paraformaldehyde and 0.05% (v/v) glutaraldehyde overnight at 4°C. DNA was visualised with 1 μg/ml DAPI immediately before analysis.

3D-Imaging was done with a fully automated iMIC widefield fluorescence microscope (FEI-TILL Photonics, Munich, Germany), equipped with a Sensicam qe CCD camera (PCO, Kelheim, Germany; pixel size 6.45 μm) and a 100x oil objective (NA 1.4). Deconvolution of image stacks was performed with the Huygens Essential software (Scientific Volume Imaging B.V., Hilversum, Netherlands). Fluorescence images are shown as maximum intensity projections of 3D-stacks in false colours with green fluorescence in green and blue fluorescence in grey.

## Scanning electron microscopy

Explanted tsetse alimentary tracts were fixed in Karnovsky solution (2% formaldehyde, 2.5% glutaraldehyde in 0.1M cacodylate buffer, pH 7.4) and incubated overnight at 4°C. Samples were washed 3 times for 5 min at 4°C with 0.1 M cacodylate buffer, pH 7.4, followed by incubation for 1 hr at 4°C in post-fixation solution (2.5% glutaraldehyde in 0.1 M cacodylate buffer, pH 7.4). After additional washing, the samples were incubated for 1 hr at 4°C in 2% tannic acid in cacodylate buffer, pH 7.4, 4.2% sucrose, and washed again in water (3x for 5 min, 4°C). Finally, serial dehydration in acetone was performed, followed by critical point drying and platinum coating. Scanning electron microscopy was done using the JEOL JSM-7500F field emission scanning electron microscope (JEOL, Freising, Germany).

## Nuclear – posterior cell pole measurements

Cells were harvested and fixed as above, before being mounted directly onto slides containing 3 μl antifade mounting medium with DAPI (Vectashield, California, USA). Images were taken using a DMI6000B widefield fluorescence microscope (Leica microsystems, Mannheim, Germany), equipped with a DFC365FX camera (pixel size 6.45 μm) and a 100x oil objective (NA 1.4). Images (DIC and DAPI) were overlaid using FIJI (NIH, Bethesda, Maryland)(*Schneider et al., 2012*) and measurements taken from the centre of the cell nucleus, along the midline, to the posterior end of the cell.

## Statistics

Statistical analyses were performed using the statistical framework R vers 4.02 (*R Development Core Team, 2020*). The PAD1 and EP -positive and -negative cells were modelled by the different cell types (Stumpy or Slender), a spline for the time parameter (k=4, dimension of the spline basis) and a binomial link function. Finally, the model was fitted using the function 'gam' as implemented in the 'mgcv' package (*Wood et al., 2016*). Unpaired, parametric t-tests were performed using Graphpad prism version 8.4.0 for macOS (Graphpad software, San Diego, California USA, http://www.graphpad.com).

## Acknowledgements

We thank Nicola Jones, Susanne Kramer, Manfred Alsheimer, Christian Janzen and Ricardo Benavente for discussion and critical reading of the manuscript. We thank Thomas Müller for help with data presentation using python. We thank Alyssa Borges for fruitful discussions about statistics. We thank Keith Matthews (Edinburgh) for the anti-PAD1 antibody. BM is supported by DFG grant number 396187369. ME is supported by DFG grants EN305, SPP1726 (Microswimmers – From Single Particle Motion to Collective Behaviour), GIF grant I-473–416.13/2018 (Effect of extracellular *Trypanosoma brucei* vesicles on collective and social parasite motility and development in the tsetse fly), GRK2157 (3D Tissue Models to Study Microbial Infections by Obligate Human Pathogens), and NUM Organostrat (Bundesministerium für Bildung und Forschung). ME is a member of the Wilhelm Conrad Roentgen Center for Complex Material Systems (RCCM).

## Additional information

### Funding

| Funder | Grant reference number | Author |
|---|---|---|
| Deutsche Forschungsge-meinschaft | EN305 | Markus Engstler |
| Deutsche Forschungsge-meinschaft | SPP1726 | Markus Engstler |
| German-Israeli Foundation for Scientific Research and Development | ant I-473-416.13/2018 | Markus Engstler |
| Deutsche Forschungsge-meinschaft | GRK2157 | Markus Engstler |
| Deutsche Forschungsge-meinschaft | 396187369 | Brooke Morriswood |
| Bundesministerium für Bildung und Forschung | NUM Organostrat | Markus Engstler |

The funders had no role in study design, data collection and interpretation, or the decision to submit the work for publication.

### Author contributions

Sarah Schuster, Conceptualization, Data curation, Formal analysis, Validation, Investigation, Visualization, Methodology, Writing - original draft; Jaime Lisack, Data curation, Formal analysis, Validation, Investigation, Visualization, Methodology, Writing - original draft, Writing - review and editing; Ines Subota, Conceptualization, Resources, Formal analysis, Supervision, Investigation, Visualization, Methodology; Henriette Zimmermann, Validation, Investigation, Visualization, Methodology; Christian Reuter, Investigation, Methodology; Tobias Mueller, Software, Formal analysis, Validation, Methodology; Brooke Morriswood, Data curation, Validation, Visualization, Methodology, Writing - original draft, Writing - review and editing; Markus Engstler, Conceptualization, Resources, Data curation, Supervision, Funding acquisition, Validation, Investigation, Methodology, Writing - original draft, Project administration, Writing - review and editing

### Author ORCIDs

Jaime Lisack (iD) https://orcid.org/0000-0001-9621-4000
Brooke Morriswood (iD) https://orcid.org/0000-0001-7031-3801
Markus Engstler (iD) https://orcid.org/0000-0003-1436-5759

### Decision letter and Author response

Decision letter https://doi.org/10.7554/eLife.66028.sa1
Author response https://doi.org/10.7554/eLife.66028.sa2

## Additional files

### Supplementary files

• Transparent reporting form

### Data availability

All original data are in the submission.

The following dataset was generated:

| Author(s) | Year | Dataset title | Dataset URL | Database and Identifier |
|---|---|---|---|---|
| Tamburrino G, | 2020 | Molecular dynamics simulation | https://figshare.com/arti- | figshare, 10.6084/m9. |

| Zachariae U | trajectories, AmtB in twin-His HSD-HSE and HSE-HSD states | cles/dataset/Molecular_dynamics_simulation_trajectories_AmtB_in_twin-His_HSD-HSE_and_HSE-HSD_states/12826316 | figshare.12826316 |

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

## Appendix 1

Information on the number of trypanosomes in human blood samples, based on original observations reported by Robert Koch in 1906/07.

### (Translated with DeepL Pro)
### Robert Koch. Report on the activities of the commission sent to East Africa to research sleeping sickness in 1906/07 (Verlag von Julius Springer, Berlin, 1909)

(page 17) "If now sick people, in whose blood trypanosomes can be detected, are examined quite carefully and daily, as we have done several times, then one first learns that the number of trypanosomes in the blood is almost always very low. Often only one or two trypanosomes are present in a preparation containing several drops of blood. Five to ten trypanosomes in one preparation are already a rather rich yield. We have only exceptionally seen a larger number of trypanosomes, so that every second to third field of view of the very thick layer of the preparation was filled with one trypanosome. Such quantities of trypanosomes, which are almost regularly seen in the blood of laboratory animals, have never been found in the blood of humans.

The occurrence of trypanosomes in the blood is quite irregular. If they were found for one or a few days, they suddenly disappear and usually stay away for 2 to 3 weeks, only to reappear again.

They are then very sporadic at the beginning, become a little more numerous the next day and maybe even the third day, then decrease again for one or two days and disappear again. It seems as if they appear periodically in the blood, their presence lasts 2 to 5 days and their absence 2 to 3 weeks. In most cases, the recurrence of trypanosomes is associated with an increase in temperature and increased symptoms of disease, especially headaches and chest pain.

It is necessary to be familiar with the periodic appearance of trypanosomes in the blood in order not to make too many futile examinations during the diagnostic examination of the blood.

In blood preparations, trypanosomes have a very different appearance depending on whether they lie on the band or more towards the inside. On the periphery they appear in terms of their size, the shape of the nucleus, visibility of the undulating membrane and the flagella, just as one is used to see them in smear preparations of the blood of the test animals. But in the thick layers of the inner parts of the preparation they look considerably smaller, their colour is darker, they also have a rounded appearance, the nucleus is smaller, membrane and flagellum are hardly visible, often they seem to be missing. However, this different appearance is not due to the different composition of the trypanosomes, but is only caused by the preparation. At the edges they dry up in a very thin layer and very fast. They are thus spread out, stretched to a certain extent and immediately fixed in this form by drying. In the thick blood layer of the preparation, the drying process is only gradual, leaving the Trypanosoma time to dry in its original cylindrical shape with more or less strong shrinking of the whole body and especially of the undulating membrane and the flagella.'

### (Original Text in German)
### Robert Koch. Bericht über die Tätigkeit der zur Erforschung der Schlafkrankheit im Jahre 1906/07 nach Ostafrika entsandten Kommission (Verlag von Julius Springer, Berlin, 1909)

(Seite 17) „Wenn nun Kranke, in deren Blut Trypanosomen nachzuweisen sind, recht sorgfältig und täglich untersucht werden, wie wir das des öfteren getan haben, dann erfährt man zunächst, daß die Anzahl der Trypanosomen im Blute fast immer eine sehr geringe ist. Auf ein Präparat, welches mehrere Tropfen Blut enthält, kommen oft nur ein oder zwei Trypanosomen. Fünf bis zehn Trypanosomen in einem Präparat bilden schon eine ziemlich reiche Ausbeute. Wir haben nur ausnahmsweise eine größere Zahl von Trypanosomen gesehen, so daß auf jedes zweite bis dritte Gesichtsfeld der sehr dicken Präparatenschicht ein Trypanosoma kam. Solche Mengen von Trypanosomen, wie man sie fast regelmäßig im Blute der Versuchstiere zu sehen bekommt, haben wir niemals im Blute der Menschen angetroffen.

Das Vorkommen der Trypanosomen im Blute ist ziemlich unregelmäßig. Wenn sie einen oder einige Tage lang gefunden wurden, dann sind sie plötzlich verschwunden und bleiben gewöhnlich 2 bis 3 Wochen fort, um dann wieder zum Vorschein zu kommen.

Sie sind dann anfangs ganz vereinzelt, werden am nächsten und vielleicht auch noch am dritten Tage ein wenig zahlreicher, nehmen dann wiederum ein bis zwei Tage ab und verschwinden von neuem. Es hat den Anschein, als ob sie periodenweise im Blute erscheinen, und zwar dauert ihr Vorhandensein 2 bis 5 Tage und ihr Fehlen 2 bis 3 Wochen. Meistens sind mit dem Wiederauftreten der Trypanosomen eine Temperatursteigerung und verstärkte Krankheitssymptome, namentlich Kopf- und Brustschmerzen, verbunden.

Man muß mit dem periodenweisen Erscheinen der Trypanosomen im Blute vertraut sein, um bei der diagnostischen Untersuchung des Blutes nicht zu viele vergebliche Untersuchungen zu machen.

In den Blutpräparaten haben die Trypanosomen ein sehr verschiedenes Aussehen, je nachdem sie am Bande oder mehr nach dem Innern zu liegen. Am Rande erscheinen sie in bezug auf ihre Größe, auf die Gestalt des Kerns, Sichtbarkeit der undulierenden Membran und der Geißel, ebenso wie man sie in Ausstrichpräparaten vom Blut der Versuchstiere zu sehen gewohnt ist. Aber in den dicken Schichten der inneren Partien des Präparates sehen sie erheblich kleiner aus, ihre Farbe ist dunkler, sie haben auch ein rundliches Aussehen, der Kern ist kleiner, Membran und Geißel sind kaum zu erkennen, oft scheinen sie zu fehlen. Dieses verschiedene Aussehen beruht nun aber nicht auf verschiedener Beschaffenheit der Trypanosomen, sondern ist nur durch die Präparation bedingt. Am Rande trocknen sie in sehr dünner Schicht und sehr schnell ein. Dabei werden sie also der Fläche nach ausgebreitet, gewissermaßen gestreckt und in dieser Form durch das Eintrocknen sofort fixiert. In der dicken Blutschicht des Präparats geht der Eintrocknungsprozeß nur allmählich vor sich, und da bleibt dem Trypanosoma Zeit, in seiner ursprünglichen walzenförmigen Gestalt unter mehr oder weniger starkem Schrumpfen des ganzen Körpers und ganz besonders der undulierenden Membran und der Geißel zu trocknen.'

## Based on Koch's observations, the following estimations can be made:

One drop of blood = **50** µl and 'several drops' are five drops = 250 µl; maximum count was 5–10 trypanosomes per five drops on average, which means 20–40 trypanosomes are present in one milliliter of blood. Hence, one tsetse bloodmeal of 20 µl would contain 0.4 to 0.8 trypanosomes.

One drop of blood = **20** µl and 'several drops' are five drops = 100 µl; maximum count was 5–10 trypanosomes per five drops on average, which means 50–100 trypanosomes per ml are present in one milliliter of blood. Hence, one tsetse bloodmeal of 20 µl would contain 1 to 2 trypanosomes.

