## [Decision Letter]

**Acceptance summary:**

*Trypanosoma brucei* lives as "bloodstream forms" in the blood and tissue fluids of mammals, where they divide as long slender forms which are dependent on substrate-level phosphorylation during glycolysis. Meanwhile, in the Tsetse fly vector midgut, they divide as "procyclic forms", which rely heavily on mitochondrial metabolism. While in the blood, the parasites differentiate – probably via quorum sensing – to a non-dividing "stumpy form" which has gene expression changes that pre-adapt the parasites to life in the fly. Textbook knowledge is that only the stumpy form can differentiate to the procyclic form. This paper overturns this dogma. The authors show that long slender forms can also differentiate – somewhat more slowly – directly into procyclic forms in the fly, by-passing several known aspects of stumpy formation.

**Decision letter after peer review:**

Thank you for submitting your article "Unexpected plasticity in the life cycle of *Trypanosoma brucei*" for consideration by *eLife*. Your article has been reviewed by 4 peer reviewers, including Christine Clayton as the Reviewing Editor and Reviewer #1, and the evaluation has been overseen by Miles Davenport as the Senior Editor. The following individual involved in review of your submission has agreed to reveal their identity: James Morris (Reviewer #3).

Essential revisions:

Your paper has been reviewed by the reviewing editor and 3 other reviewers, of whom one was less enthusiastic than the other two. Overall it was agreed that:

1. More statistical analysis is essential, everywhere. At present there is very little indeed. This could partly be achieved by presenting the data as box plots (or "beeswarm" plots) using data from individual flies. From Table 1, it is important to display not just the transmission index (as in Figure 2), but also the individual values for Tsetse fly infection (MG, PV, SG) with the 2-parasite inoculum (lines iii, vi, viii, xi and xii). These are quite complicated to compare from the Table and there are clear differences. Similarly for the bar graphs, it is essential overlay with the values obtained in the individual replicates as dots (not SD!). Readers must be able to assess the variability in ALL of the measurements. One referee also suggested replacing the Table in Figure 4 with more plots – the proportions of different N-K conformations could be shown within the PAD columns, or separately as much narrower columns. The actual numbers could then go into the Supplement.

2. The GFP-positive cells should be described as GFP-positive – not PAD1-positive. The caveats for this assay should be mentioned.

3. Some quantitative analysis of cell morphology (e.g. K-N distances) beyond the GFP should be done – this can almost certainly be done using existing images.

4. One new experiment is needed – assessing GFP-PAD expression in the inoculum after the filtration steps to get rid of the methyl cellulose. (Or is this already what was done?)

5. Can the metacyclics be tested for VSG expression? As an alternative I guess you could use morphology and state that strictly speaking, metacyclic identity has not been confirmed. The rabbit VSG117 antibody that my lab has seems to be fairly non-specific but maybe you have anti Antat1.1? Another option would be to stain for PGK. (Though I actually don't know where PGK is in epimastigotes.) This would be nice but is probably not essential.

Further suggestions for improvement are in the individual reviews.

I would also add that there is a really nice new review about stumpies which is highly relevant and warrants citation: Trends Parasitol. 2020 Dec 11:S1471-4922(20)30316-0. doi: 10.1016/j.pt.2020.11.003.

I also seem to remember that Muriel Roberts originally thought that it was the slender forms that were transmissible, but I may be wrong.

*Reviewer #1 (Recommendations for the authors):*

I agree with the authors that the data in this paper indicate that slender forms can productively infect Tsetse. Although their initial differentiation is delayed, they ultimately are as successful as stumpy forms. One could add that although stumpy morphology is not an essential intermediate, stumpy-like gene expression may well be: essentially, conditions in the fly are sufficient to induce the developmental programme. I don't think the data support the conclusion that no cell-cycle arrest is required – although such an arrest does have to be quite brief.

I have one major criticism, and that is the authors' equation of GFP-positive with "PAD1 positive". Yes, this means that the GFP-PAD1 mRNA has been present. But the GFP could easily persist after the mRNA has gone, and of course the presence of GFP can't be equated with the presence of PAD1 protein. This limitation has to be stated right at the start and the description re-worded throughout. Has to be something like "GFP-PAD reporter positive", or maybe even just "GFP-positive".

Line 80 – The possibility that densities are higher in the skin should be mentioned briefly (I know it's in the Discussion as well.)

Line 99 – for non-experts, the authors should mention that developmental regulation of PAD1 is controlled by the 3'-UTR.

Line 132 and Table 1: the authors should, in the supplement, provide a simple plot showing the probability distribution for numbers of parasites in 20µl at the different parasite densities. The Legend (as well as the text) should briefly define transmission index and the number of flies per group should be stated. Also, say what the different groups are: what are slender ES and slender SIF? How long was ES induced and what exactly are the "SIF" stumpies? (I know it's in the Methods section but the density of the cultures should be given.) Statistical analysis is needed. Are any of the differences really significant?

Video 2A – expand the legends, these were initially difficult to follow. (Just say that they switch from DIC to fluorescence, and which comes first.)

Line 162 – this is a bit misleading. The authors should define what they mean by "infection efficiency"? Indeed the transmission indices for slender and stumpy are similar, but the number of infections per input slender trypanosome is half of that obtained with stumpy. The should state up front (rather than much later) that the slender parasites have more trouble establishing infection in the midgut.

Line 199-120 – a bit over-stated. If there is no midgut infection then there will definitely be no tryps in the salivary glands.

Figure 5 and Figure 7 – titles are not appropriate since cell cycle arrest was not measured. How about "Slender trypanosomes show delayed EP procyclin expression after Tsetse infection / in vitro differentiation"? Please explain why 2h time-ranges are given rather than single times.

I do not think it is possible to conclude that absolutely no cell-cycle arrest is required. From Figure 6, can a 1-2 h arrest while PAD1 is turned on and the rest of the programme is initiated be excluded? I don't think so – the temporal definition and numbers simply don't allow it. Is the sudden drop in positive 2K1N cells from the slender cells at 15-17h significant?

Line 254 – Did the authors stain for GPEET or phosphorylated GPEET, which are often expressed first after differentiation? IF not they should surely mention that the procyclic-morphology cells are most likely expressing GPEET.

Line 292 – clearly not true since the procyclin expression of slender forms was delayed. Also no other markers of procyclics were measured. (From their existing images can authors also include the N-K and N-posterior tip distances of 1K1N cells, which gives procyclic morphology independently of procyclin expression?) For Figures 6 and 7, please also plot the absolute cell densities in addition to showing the percentages.

Discussion

"We have not observed cell death of monomorphic or pleomorphic slender cells in infected tsetse midguts." But how would you detect it?

The authors emphasise that the slender-derived parasites are "more successful in the second part of the life cycle." Are the differences in Table 1, which are reiterated in lines 300-302, really statistically significant?

*Reviewer #3 (Recommendations for the authors):*

1. The data as presented in the tables is challenging to follow. In table 1, I recommend moving data on short stumpy and slender form related to expression site attenuation to supplemental data. In Tables 2 and 6, the information in the bars of the bar graphs (the different dotted lines, for example) should be developed into stand-alone parts of the figure, as they are difficult to interpret. In the text of the results, it would be helpful to be consistent with directing the reader to the correct column (for example, Line 192, "column 5).

2. It is unclear what tissues were analyzed in Figure 4 – some clarification about what an "explant" is needed.

*Reviewer #4 (Recommendations for the authors):*

This is a very important finding that challenges the current dogma. As any paper that challenges the dogma, this paper raises many questions that will be tackled in future studies, namely reproducing these results with parasites recently isolated from the wild, instead of tissue culture adapted lines. Another important avenue will be to test the different mutant parasites that are incapable of differentiating from slender to stumpy forms and dissect the molecular machinery that composes the slender-differentiation.

I describe below two points that, in my opinion, could help strengthen the paper:

1. I am concerned about the parasite purification protocol prior to Tsetse infection (which involves change of buffers, filtration to remove the methylcellulose matrix and centrifugation). Could the protocol stress the slender forms such that these non-cell-cycle-arrested-stressed slenders gain the capacity to infect Tsetses? Previous reports have shown that reduced cellular energy promotes parasite differentiation (Barquilla 2012; Saldivia, 2016). Could the authors avoid or minimize the purification steps? Some *T. brucei* pleomorphic strains grow in vitro without methylcellulose, which would simplify parasite harvesting. Could the parasites be simply diluted in blood prior to infection of flies (without any purification)? When was the IFA done to confirm expression of GFP::PAD1? Ideally, it should have been at the end of purification protocol, just prior to Tsetse feeding. If it is technically impossible to improve/avoid parasite purification protocol, could the authors provide some evidence that after purification parasites are not stressed? Would purified parasites grow exponentially in culture without a lag phase? What are the levels of phosphorylated AMPKa1? This type of experiments would help ruling out that parasites were stressed by a reduced cellular energy during purification protocol.

2. The authors showed that the kinetics of differentiation of slender and stumpy forms to procyclic forms is different both in vitro and in vivo, although both lead to the formation of bonafide procyclic forms. The authors could consider doing a competition experiment between slenders and stumpy forms to test their individual fitness in group.

[Editors' note: further revisions were suggested prior to acceptance, as described below.]

Thank you for resubmitting your work entitled "Unexpected plasticity in the life cycle of *Trypanosoma brucei*" for further consideration by *eLife*. Your revised article has been reviewed by 3 peer reviewers and the evaluation has been overseen by Miles Davenport as the Senior Editor, and a Reviewing Editor.

The manuscript has been improved but there are some remaining issues that need to be addressed, as outlined below:

Reviewer 4 asks for you to clarify one sentence and I agree. I too think that the phrase "without harvesting the cells" is really difficult to understand.

*Reviewer #2 (Recommendations for the authors):*

My only real comment is that the videos seem a little out of place. M1 of the fly feeding has little obvious value in the present context, and I found he quality of M2 disappointing. M2 could be improved and if this were done it would then be a genuine addition to the paper.

*Reviewer #3 (Recommendations for the authors):*

The authors have addressed my concerns through revision. I applaud their efforts to improve the visualization of the data while maintaining transparency, as there is a tremendous amount to interpret.

*Reviewer #4 (Recommendations for the authors):*

The authors have convincingly responded to my question about a putative "stress" effect on harvested parasites. Controls at multiple levels have been used. Thank you.

While their answer is very clear in the rebuttal, these arguments are not clearly described in the manuscript (or did I miss them?). For example, the description of one of the controls: "cells were directly taken from culture and added to blood, thereby completely omitting any filtration step.", in the rebuttal, is described in the paper (line 483) as "the dilutions made directly in blood without harvesting the cells". I suggest the authors include in the paper the exact words used in the rebuttal to answer my question.

The authors state that my second point (competition) is beyond the scope of this paper. I support the editor's decision.

---

## [Author Response]

Essential revisions:Your paper has been reviewed by the reviewing editor and 3 other reviewers, of whom one was less enthusiastic than the other two. Overall it was agreed that:1. More statistical analysis is essential, everywhere. At present there is very little indeed. This could partly be achieved by presenting the data as box plots (or "beeswarm" plots) using data from individual flies. From Table 1, it is important to display not just the transmission index (as in Figure 2), but also the individual values for Tsetse fly infection (MG, PV, SG) with the 2-parasite inoculum (lines iii, vi, viii, xi and xii). These are quite complicated to compare from the Table and there are clear differences.

For the revised version, we teamed up with an expert mathematician (Tobias Müller) who performed state-of-the-art modelling and statistical analysis of our data. For the figures with bar graphs (Figure 4-7), statistical analysis was run using a *point estimate mod*el based on time vs. cell type (stumpy and slender).

For Figure 2A, we now additionally provide a beeswarm plot with results from all individual fly infections (2B), as well as the total number of infections which go with this figure (Figure 2 – table supplement 1).

A plot showing the transmission index for each N, as well as a table with total infection numbers (rather than percent) and number of infections per N, was added to the supplements (Figure 3—figure supplement 1 and – table supplement 1).

All statistical validation supported the interpretation of the data. We hope that these steps have sufficiently addressed the reviewer's concern.

Similarly for the bar graphs, it is essential overlay with the values obtained in the individual replicates as dots (not SD!). Readers must be able to assess the variability in ALL of the measurements.

The tables of cell cycle profiles for PAD1-positive and -negative cells were moved into the supplements (Figure 5 – Supplemental Table 1 and Figure 7 – Supplemental Table 1) and replaced with graphs depicting the PAD1-positive and -negative dividing populations (Figure 5 B, C; Figure 7 B, C). They are shown as mean with standard deviation. A table showing the conversion from trypanosomes/bloodmeal to trypanosomes/ml was added as requested as Figure 2 – table supplement 1.

One referee also suggested replacing the Table in Figure 4 with more plots – the proportions of different N-K conformations could be shown within the PAD columns, or separately as much narrower columns. The actual numbers could then go into the Supplement.

Nucleus (N) – posterior pole measurements have now been conducted for all timepoints and added to the supplements. While N-kinetoplast (K) was an option, we decided for N-posterior pole so that a difference can also be seen between slender and stumpy at the zero hour timepoint (which cannot really be differentiated with N-K). Data were visualised using violin plots, with the line at the median and dotted lines at the quartiles while tables added below show some descriptive measures from the data (Figure 7—figure supplement 2).

2. The GFP-positive cells should be described as GFP-positive – not PAD1-positive. The caveats for this assay should be mentioned.

The caveats are stated in line 97-105:

“Expression of the protein associated with differentiation 1 (PAD1) is accepted as a marker for development to the stumpy stage (Dean et al., 2009). […] The validity of the GFP:PAD1^UTR^ reporter as an indicator for the activation of the PAD1 pathway has been reported previously (Batram et al., 2014; Zimmermann et al., 2017), and further corroborated by co-staining with an antibody against the PAD1 protein (Figure 1—figure supplement 1).”

We also mention this several times in the manuscript but for the sake of clarity and readability we have decided not to include the description each time the reporter has been used. We hope this is acceptable.

3. Some quantitative analysis of cell morphology (e.g. K-N distances) beyond the GFP should be done – this can almost certainly be done using existing images.

As there is no true discernable difference between stumpy and slender cell K-N distances, we chose instead to measure the distance between the nucleus and posterior pole of both slender and stumpy trypanosomes during differentiation to the procyclic insect stage – this distance is more discernable between slender, stumpy, and procyclic, especially for stumpy and slender. A violin plot showing all measurements, with lines at the median and dotted lines at the quartiles, was added to the supplements (Figure 7—figure supplement 2). Included is a table with some descriptive information about the data.

4. One new experiment is needed – assessing GFP-PAD expression in the inoculum after the filtration steps to get rid of the methyl cellulose. (Or is this already what was done?)

We have considered the possibility that the cells might be “stressed” even by a single, brief filtration step, and made sure that this did not result in increased GFP-PAD1 expression.

Almost all experiments were done using the same harvesting protocol. In Figure 2A, however, for 0.2, 1, and 2 cells/bloodmeal, cells were *directly* taken from culture and added to blood, thereby completely omitting any filtration step. This was not necessary since such a small number of cells were needed. These very low trypanosome numbers per bloodmeal still resulted in infections. Furthermore, all images taken at the start of the experiment (zero hour timepoint) were taken *after* the harvest. The cells did not express GFP-PAD1. As a further control, we regularly put harvested cells back in culture where they instantaneously resumed normal growth and did not show GFP-PAD1 expression. In conclusion, the experiments have been carefully controlled on multiple levels.

5. Can the metacyclics be tested for VSG expression? As an alternative I guess you could use morphology and state that strictly speaking, metacyclic identity has not been confirmed. The rabbit VSG117 antibody that my lab has seems to be fairly non-specific but maybe you have anti Antat1.1? Another option would be to stain for PGK. (Though I actually don't know where PGK is in epimastigotes.) This would be nice but is probably not essential.

In the course of another project, we have used the same tsetse colony and identical experimental procedures to allow infection of artificial skin equivalents with tsetse-borne metacyclic parasites. The respective manuscript is currently under review and is available on bioRxiv (https://doi.org/10.1101/2021.05.13.443986). In this work, we provide extensive time-resolved single cell RNA-seq analyses, including that of metacyclic and naïve slender parasites. While the metacyclic cells express metacyclic VSGs, the re-activated slender parasites express Antat1.1 VSG in a subset of cells, as expected. Furthermore, the vast majority of cells found freely swimming in the salivary glands after fly infection are metacyclic forms. There are a few epimastigote trypanosomes detectable, which is not so surprising given that they arrive at the salivary glands as freely swimming trypanosomes.

Further suggestions for improvement are in the individual reviews.I would also add that there is a really nice new review about stumpies which is highly relevant and warrants citation: Trends Parasitol. 2020 Dec 11:S1471-4922(20)30316-0. doi: 10.1016/j.pt.2020.11.003.

We have added this reference (which was published after our initial submission – the only reason for its initial omission). Lines 338-340 now read:

“It is also of note that a recent review has suggested that stumpy trypanosomes may have evolved as a stress response, with procyclic pathways already activated in order to increase chances of survival in the fly (Quintana et al., 2021).”

Reviewer #1 (Recommendations for the authors):I agree with the authors that the data in this paper indicate that slender forms can productively infect Tsetse. Although their initial differentiation is delayed, they ultimately are as successful as stumpy forms. One could add that although stumpy morphology is not an essential intermediate, stumpy-like gene expression may well be: essentially, conditions in the fly are sufficient to induce the developmental programme. I don't think the data support the conclusion that no cell-cycle arrest is required – although such an arrest does have to be quite brief.

Stumpy trypanosomes are arrested in G_0_, and no replicative cell cycle is detectable until either cell death, which occurs after 3 days (our own measurements), or development to procyclic cells. We in fact state that slender parasites also launch the PAD1 pathway in the tsetse fly. We have no indication for a G_0_ arrest and likewise not for a G1 prolongation (as shown for example in Batram et al., 2014 and Zimmermann et al., 2017). We are actually not aware of a *bona fide* cell cycle *arrest* that lasts for a shorter period than the cell cycle.

I have one major criticism, and that is the authors' equation of GFP-positive with "PAD1 positive". Yes, this means that the GFP-PAD1 mRNA has been present. But the GFP could easily persist after the mRNA has gone, and of course the presence of GFP can't be equated with the presence of PAD1 protein. This limitation has to be stated right at the start and the description re-worded throughout. Has to be something like "GFP-PAD reporter positive", or maybe even just "GFP-positive".

We now clearly state in line 97-105:

“Expression of the protein associated with differentiation 1 (PAD1) is accepted as a marker for development to the stumpy stage (Dean et al., 2009). […] The validity of the GFP:PAD1^UTR^ reporter as an indicator for the activation of the PAD1 pathway has been reported previously (Batram et al., 2014; Zimmermann et al., 2017), and further corroborated by co-staining with an antibody against the PAD1 protein (Figure 1—figure supplement 1).”

We also mention this several times later but for the sake of clarity and readability we have not included the description each time the reporter has been used. We hope this is an acceptable compromise.

Line 80 – The possibility that densities are higher in the skin should be mentioned briefly (I know it's in the Discussion as well.)

We now mention this in the introduction. Lines 75-79:

“Although trypanosomes might be found in higher densities in the skin (Capewell et al., 2016), chronic trypanosome infections are characterised by low blood parasitemia, meaning that the chance of a tsetse fly ingesting any trypanosomes, let alone short-lived stumpy ones, is also very low (Frezil, 1971; Wombou Toukam et al., 2011).”

The referee might also consider our recent new work on fly infections of artificial skin, which clearly shows that immediately after fly infection, trypanosomes do enter a dormant stage (https://doi.org/10.1101/2021.05.13.443986).

Line 99 – for non-experts, the authors should mention that developmental regulation of PAD1 is controlled by the 3'-UTR.

This is a good point and was added in line 99-105:

“As the 3’UTR of the PAD1 gene regulates the expression of pad1 (MacGregor and Matthews, 2012), cells expressing an NLS-GFP reporter fused to the 3' UTR of the PAD1 gene (GFP:PAD1^UTR^) will have GFP-positive nuclei when the PAD1 gene is active. […] pathway has been reported previously (Batram et al., 2014; Zimmermann et al., 2017), and further corroborated by co-staining with an antibody against the PAD1 protein (Figure 1—figure supplement 1).”

Line 132 and Table 1: the authors should, in the supplement, provide a simple plot showing the probability distribution for numbers of parasites in 20µl at the different parasite densities.

Instead of a probability plot, we have given a supplemental table with the calculation of the probable number of trypanosomes per 20µl.

The Legend (as well as the text) should briefly define transmission index and the number of flies per group should be stated.

Done. See Lines 585-586 and Figure 2 – table supplement 2.

Also, say what the different groups are: what are slender ES and slender SIF? How long was ES induced and what exactly are the "SIF" stumpies? (I know it's in the Methods section but the density of the cultures should be given.)

Done. See lines 577-583.

Statistical analysis is needed. Are any of the differences really significant?

For the revision, we teamed up with an expert mathematician (Tobias Müller) who performed state-of-the-art modelling and statistical analyses of our data. For the figures with bar graphs (Figure 4-7), statistical analysis was run using a point estimate model based on time vs. cell type (stumpy and slender), including both confidence intervals (seen as transparent colour) and the individual replicates for each experiment (seen as dots).

For Figure 2A, we now additionally provide a beeswarm plot with results from all individual fly infections (2B), as well as the total number of infections which go with this figure (Figure 2 – table supplement 1).

A plot showing transmission index for each N, as well as a table with total infection numbers (rather than percent) and number of infections per N, was added to the supplements (Figure 3—figure supplement 1 and – table supplement 1).

Video 2A – expand the legends, these were initially difficult to follow. (Just say that they switch from DIC to fluorescence, and which comes first.)

Done. See lines 741-742.

Line 162 – this is a bit misleading. The authors should define what they mean by "infection efficiency"? Indeed the transmission indices for slender and stumpy are similar, but the number of infections per input slender trypanosome is half of that obtained with stumpy. The should state up front (rather than much later) that the slender parasites have more trouble establishing infection in the midgut.

We now state:

“The infection efficiency, using TI as a measure, when the flies were fed with either 20 stumpy trypanosomes or 20 pleomorphic slender trypanosomes was similar (Figure 2A, compare TI in column 8 for rows ii and vii). […] This TI of 0.60 was identical for both populations of slender cells (Figure 3)”.

Furthermore, the abundance of slender trypanosomes, when compared to the short-lived stumpy trypanosomes, should be more than 2-fold higher. This would compensate for any initial delay in transformation to the procyclic stage. As we know that this is rather speculative, we do not discuss this point in the manuscript.

Line 199-120 – a bit over-stated. If there is no midgut infection then there will definitely be no tryps in the salivary glands.

We agree with the referee.

Figure 5 and Figure 7 – titles are not appropriate since cell cycle arrest was not measured.

We agree and have changed the title accordingly. The title for figure 5 (now figure 6) reads “Slender trypanosomes show delayed expression of EP compared to stumpy trypanosomes, while they seamlessly differentiate to the procyclic life cycle stage in the tsetse fly.” The title for figure 7 (now figure 8) reads “Slender trypanosomes show delayed expression of EP while they differentiate to the procyclic life cycle stage in vitro.”

How about "Slender trypanosomes show delayed EP procyclin expression after Tsetse infection / in vitro differentiation"? Please explain why 2h time-ranges are given rather than single times.

Two-hour time-ranges were used to account for the time required for fly dissection and examining the explanted organs. Hence, we did not want to imply more accuracy than the experiment offered. Nevertheless, we now give the start time of the interval and state this in the legend at lines 613-614 and 640-641.

I do not think it is possible to conclude that absolutely no cell-cycle arrest is required.

The conclusion that there is no cell cycle arrest in fact cannot be drawn in our paper, that is why we now instead refer to ‘no apparent cell cycle arrest’. Please also see our comment above.

From Figure 6, can a 1-2 h arrest while PAD1 is turned on and the rest of the programme is initiated be excluded? I don't think so – the temporal definition and numbers simply don't allow it.

See above. A 1-2 hours stalling of the cell cycle cannot be regarded as a cell cycle arrest (G_0_), but rather prolongation of the cell cycle phase (maybe in G_1_). We are not aware of any example in which bona fide cell cycle arrest was shorter than the cell cycle.

Is the sudden drop in positive 2K1N cells from the slender cells at 15-17h significant?

We now give graphs showing PAD1-positive and -negative cells throughout the time course and the answer is no. It is within the standard deviation of the neighboring points.

Line 254 – Did the authors stain for GPEET or phosphorylated GPEET, which are often expressed first after differentiation? IF not they should surely mention that the procyclic-morphology cells are most likely expressing GPEET.

Lines 261-265 now read:

“GPEET is another procyclic surface protein that is expressed early in the transition from bloodstream stage to procyclic stage cells in the tsetse midgut, before being replaced by EP (Vassella et al., 2000). Whether these early and morphologically procyclic cells expressed GPEET was not checked, and remains a target of future work.”

Line 292 – clearly not true since the procyclin expression of slender forms was delayed.

We agree and have omitted the mention of differentiation kinetics. The line now reads:

“Furthermore, the overall developmental capacity of both life cycle stages is comparable, in vitro and in vivo.”

Also no other markers of procyclics were measured. (From their existing images can authors also include the N-K and N-posterior tip distances of 1K1N cells, which gives procyclic morphology independently of procyclin expression?) For Figures 6 and 7, please also plot the absolute cell densities in addition to showing the percentages.

As there is no true discernable difference between stumpy and slender cell K-N distances, we chose to measure K-cell posterior of both slender and stumpy transitioning to procyclic – this distance is more discernable between slender, stumpy, and procyclic, especially for stumpy and slender. A violin plot showing all measurements, with lines at the median and dotted lines at the quartiles, was added to the supplements (Figure 7—figure supplement 2). Included is a table with some descriptive information about the data.

Discussion"We have not observed cell death of monomorphic or pleomorphic slender cells in infected tsetse midguts." But how would you detect it?

The referee is right in stating that one cannot easily detect dead trypanosomes in the midguts, thus we did it the opposite way. We counted the number of live trypanosomes at early timepoints of midgut infections. The data is now given as Figure 5—figure supplement 1.

The authors emphasise that the slender-derived parasites are "more successful in the second part of the life cycle." Are the differences in Table 1, which are reiterated in lines 300-302, really statistically significant?

The transmission index suggests that in fact slender and stumpy trypanosomes complete the passage through the fly with almost equal success rates. Why one or the other life cycle stage should be successful in early or late stages of infection is a matter of speculation, and in the discussion lines 318 ff. we suggest one possibility, namely the documented differences in motile behavior of trypanosomes. We have clearly marked this as a theory.

Reviewer #3 (Recommendations for the authors):1. The data as presented in the tables is challenging to follow. In table 1, I recommend moving data on short stumpy and slender form related to expression site attenuation to supplemental data. In Tables 2 and 6, the information in the bars of the bar graphs (the different dotted lines, for example) should be developed into stand-alone parts of the figure, as they are difficult to interpret. In the text of the results, it would be helpful to be consistent with directing the reader to the correct column (for example, Line 192, "column 5).

We agree with referee number 3, although we had tried our best to present the data as interpretable as possible. As this was obviously not entirely successful, we now have added a beeswarm plot (Figure 2B) so that Figure 2A is easier to navigate.

We have re-worked most figures entirely in the revised version, thereby also implementing the suggestion to extract the dotted lines and put them in separate graphs (now Figure 5 B, C and Figure 7 B, C). We also included mathematical modelling and statistical analysis to figures 5-8. We hope that this has further improved the readability of our paper.

2. It is unclear what tissues were analyzed in Figure 4 – some clarification about what an "explant" is needed.

Done. We added that it is an “explanted tsetse midgut” to the legend.

Reviewer #4 (Recommendations for the authors):This is a very important finding that challenges the current dogma. As any paper that challenges the dogma, this paper raises many questions that will be tackled in future studies, namely reproducing these results with parasites recently isolated from the wild, instead of tissue culture adapted lines. Another important avenue will be to test the different mutant parasites that are incapable of differentiating from slender to stumpy forms and dissect the molecular machinery that composes the slender-differentiation.I describe below two points that, in my opinion, could help strengthen the paper:1. I am concerned about the parasite purification protocol prior to Tsetse infection (which involves change of buffers, filtration to remove the methylcellulose matrix and centrifugation). Could the protocol stress the slender forms such that these non-cell-cycle-arrested-stressed slenders gain the capacity to infect Tsetses? Previous reports have shown that reduced cellular energy promotes parasite differentiation (Barquilla 2012; Saldivia, 2016). Could the authors avoid or minimize the purification steps? Some *T. brucei* pleomorphic strains grow in vitro without methylcellulose, which would simplify parasite harvesting. Could the parasites be simply diluted in blood prior to infection of flies (without any purification)? When was the IFA done to confirm expression of GFP::PAD1? Ideally, it should have been at the end of purification protocol, just prior to Tsetse feeding. If it is technically impossible to improve/avoid parasite purification protocol, could the authors provide some evidence that after purification parasites are not stressed? Would purified parasites grow exponentially in culture without a lag phase? What are the levels of phosphorylated AMPKa1? This type of experiments would help ruling out that parasites were stressed by a reduced cellular energy during purification protocol.

We have considered that the cells might be “stressed”, even by a single, brief filtration step, and made sure that this did not result in increased GFP-PAD1 expression.

Almost all experiments were done with the same harvesting protocol. In Figure 2A, however, for 0.2, 1, and 2 cells/bloodmeal, cells were directly taken from culture and added to blood, thereby completely omitting any filtration step. This was not necessary since it was such a small number of cells that was needed. These very low trypanosome numbers per bloodmeal still resulted in infections. Further, all images taken at the start of the experiment (zero hour timepoint) were taken *after* the harvest. The cells did not express GFP:PAD1. As a further control, we regularly put harvested cells back in culture where they instantaneously resumed normal growth and did not show GFP:PAD1 expression. In conclusion, the experiments have been carefully controlled on multiple levels.

2. The authors showed that the kinetics of differentiation of slender and stumpy forms to procyclic forms is different both in vitro and in vivo, although both lead to the formation of bonafide procyclic forms. The authors could consider doing a competition experiment between slenders and stumpy forms to test their individual fitness in group.

These experiments will be done in a follow-up paper. The experimental design is actually more complex than it might appear. For example, we have to tag the parasites with different fluorescent markers, which might differentially influence the parasites at any stage during the 30-day transition to the salivary glands.

[Editors' note: further revisions were suggested prior to acceptance, as described below.]

The manuscript has been improved but there are some remaining issues that need to be addressed, as outlined below:Reviewer 4 asks for you to clarify one sentence and I agree. I too think that the phrase "without harvesting the cells" is really difficult to understand.

We agree with the reviewer. We have now instead put: "Almost all experiments were done with the same harvesting protocol. In Figure 2A, however, for 0.2, 1, and 2 cells/bloodmeal, cells were directly taken from culture and added to blood, thereby completely omitting any filtration step. This was not necessary since it was such a small number of cells that was needed. These very low trypanosome numbers per bloodmeal still resulted in infections."

Reviewer #2 (Recommendations for the authors):My only real comment is that the videos seem a little out of place. Figure 1—video 1 of the fly feeding has little obvious value in the present context, and I found he quality of Figure 4—video 1 disappointing. Figure 4—video 1 could be improved and if this were done it would then be a genuine addition to the paper.

Reviewer #2 asks us to improve the quality of Figure 4—video 1. Since this video shows live fluorescent trypanosomes in explants of tsetse flies, the quality is as good as it gets, and the video clearly supports our conclusions. In addition, the reviewer feels that Figure 1—video 1 is out of place. We would like to keep it, as tsetse membrane feeding may be known in a few specialized trypanosome laboratories, but certainly not by the wide readership that *eLife* attracts. Therefore, it seems well placed from an educational point of view.